# `ElicitR`: Unlocking Latent Reasoning in Dense Retrievers via Generative Regularization

**Fengyu Cai** [1]  **Iryna Gurevych** [1]  **Heinz Koeppl** [1]

## Abstract

Reasoning-intensive retrieval is increasingly important for downstream applications, requiring more than lexical overlap or coarse semantic matching. While prior work mainly relies on Language Models (LMs) to synthesize reasoning-oriented supervision, we posit that it is already latent in LM-based retrievers but suppressed by contrastive overfitting. To elicit this latent reasoning, we introduce `ElicitR`, a retriever–LM framework with *generative regularization* that captures nuanced relationships among a query and its candidate documents beyond binary relevance. Concretely, alongside contrastive learning, we regularize the retriever by co-training a small LM on query–positive–negative batches. Next token prediction (NTP) for each text is conditioned on its prefix and the other in-batch texts, with cross-text conditioning weighted by retriever-computed similarities. Using MS MARCO as the *only* paired query-document supervision and a *135M* LM for generative regularization with unlabeled raw-text initialization, `ElicitR` consistently improves BRIGHT by 16-29% relative across 0.1B–3B retriever scales while maintaining performance on BEIR. At 3B, `ElicitR` reaches an nDCG@10 of 23.1, substantially outperforming larger models trained with far more curated pairs and proprietary APIs. Further analyses show that `ElicitR` prevents overfitting, improves retrieval calibration, and remains robust to batch sizes, supporting its practicality. Code is available at https://github.com/TRUMANCFY/ElicitR-ICML26.

[1]Technical University of Darmstadt, Darmstadt, Germany. Correspondence to: Fengyu Cai <fengyu.cai@tu-darmstadt.de>, Heinz Koeppl <heinz.koeppl@tu-darmstadt.de>.

*Proceedings of the $43^{rd}$ International Conference on Machine Learning*, Seoul, South Korea. PMLR 306, 2026. Copyright 2026 by the author(s).

## 1. Introduction

Retrieval-augmented generation (RAG; Lewis et al., 2020) has been widely used in knowledge-seeking and domain-specific tasks, where the retriever plays a central role in fetching relevant documents. However, beyond simple lexical or direct semantic matching, real-world queries can be complex, and identifying the relevant documents often requires intensive reasoning (Su et al., 2025). Recent work often follows the conventional paradigm, attempting to bridge the gap by using language models (LMs; Grattafiori et al., 2024) to synthesize reasoning-intensive query-document pairs (Shao et al., 2025; Das et al., 2025). However, this approach typically depends on pipeline design and quality control over the synthesis process, which may incur non-trivial cost for human validation.

Taking a step back, we ask whether we can *avoid* synthetic supervision altogether. Since modern retrievers are typically initialized with LM backbones, the complex reasoning patterns may already be latent in the model parameters (Hao et al., 2025; Zhang et al., 2025a). These capabilities may be better elicited through training rather than reintroduced via synthetic supervision. As shown in Figure 1a, during contrastive fine-tuning on MS MARCO, we observe a distinct early-peak phenomenon on a reasoning-intensive benchmark BRIGHT (Su et al., 2025), where performance maximizes mid-training before declining. In contrast, the performance on general-domain retrieval, BEIR (Thakur et al., 2021), does not exhibit this drop. This suggests that contrastive fine-tuning can overfit to binary relevance and suppress reasoning-related representation. We therefore pursue a *novel* and *orthogonal* alternative to data synthesis that mitigates this overfitting to unlock the retriever's latent reasoning capacity.

In this paper, we introduce `ElicitR`, a retriever-LM co-training framework that augments contrastive learning with *generative regularization*. Instead of relying solely on binary query-document supervision, `ElicitR` models semantic interactions among all texts within a query–positive–negative batch. Concretely, we train the LM following the NTP objective where generation is conditioned on both the current prefix and the other texts in the batch, encouraging the retriever to encode fine-grained

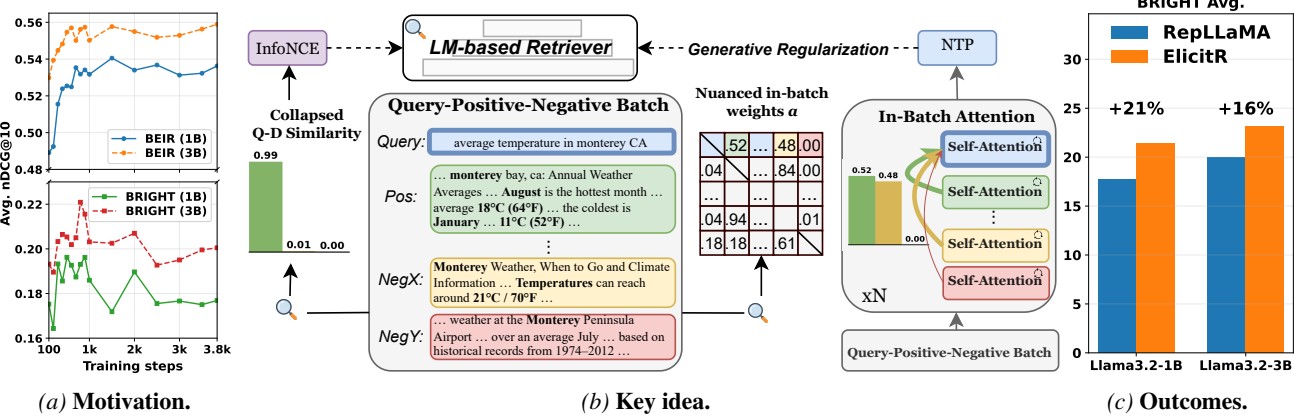

*(a)* **Motivation.**  *(b)* **Key idea.**  *(c)* **Outcomes.**

*Figure 1.* **Motivation, mechanism, and outcomes of ElicitR.** ElicitR mitigates contrastive overfitting, which degrades reasoning-intensive retrieval performance, by augmenting contrastive learning with a generative regularizer. Specifically, the retriever produces in-batch weights $\alpha$ that control cross-text attention for NTP, which back propagates to refine similarity structure. As a result, ElicitR improves performance on BRIGHT while preserving general-domain retrieval on BEIR.

dependencies beyond binary labels. As shown in Figure 1b, this objective is retriever-aware: the retriever computes similarity signals that modulate cross-text conditioning, and NTP gradients jointly update the retriever to better align with these nuanced semantic relationships. We implement this regularizer with a Revela-style in-batch attention mechanism (Cai et al., 2025), but apply it as an auxiliary objective alongside supervised contrastive training.

Empirically, ElicitR substantially improves reasoning-intensive retrieval while preserving general-domain performance. We train retrievers at four scales (0.1B–3B) by co-initializing with a small 135M LM pre-trained on raw text and fine-tuning on MS MARCO. Across all scales, ElicitR consistently outperforms the MS MARCO–finetuned contrastive baseline RepLLaMA (Ma et al., 2024) on BRIGHT by 16-29%, without sacrificing performance on BEIR (Sec.4). At the 3B scale (Figure 1c), despite using far fewer parameters, ElicitR reaches 23.1 nDCG@10 on BRIGHT, surpassing retrievers with over 7B parameters and proprietary APIs. Crucially, these gains are achieved *without* synthetic reasoning supervision. Further analyses and ablations validate the key design choices (Sec.5.1) and show that generative regularization mitigates contrastive overfitting (Sec.5.2), improves calibration (Sec.5.3 & 5.4), and remains practical under small batch sizes (Sec.5.5). To this end, we summarize our contribution as follows:

- We identify contrastive overfitting as a key obstacle in reasoning-intensive retrieval and address it with a novel solution, ElicitR, a retriever–LM framework that *elicits* latent reasoning by augmenting contrastive learning with generative regularization.
- ElicitR applies retriever-weighted in-batch NTP to query-positive-negative batches; NTP conditions on the prefix and other batch texts, and its gradients regularize

the retriever beyond binary-label supervision.
- Using MS MARCO as *only* paired query-document supervision and a 135M LM for generative regularization with unlabeled raw-text initialization, ElicitR improves BRIGHT by 16–29% across 0.1B-3B retrievers (23.1 nDCG@10 at 3B) while maintaining BEIR performance; it also mitigates contrastive overfitting, improves calibration, and is robust to small batch sizes.

## 2. Related Works

**LM Backbones for Dense Retrieval** LMs pre-trained on massive, heterogeneous corpora encode rich semantics, multi-modal, and instruction-following behavior, making them attractive backbones for dense retrievers beyond shallow encoder-only architectures (Karpukhin et al., 2020). Finetuned on advanced LM backbones, modern embedding models such as Qwen3-Embedding (Zhang et al., 2025b) and CodeXEmbed (Liu et al., 2025) achieve strong performance on multilingual retrieval and code search. Meng et al. (2025) further fine-tune multimodal LMs (Wang et al., 2024b) to obtain universal embeddings across the modalities of text, images and video. Promptriever (Weller et al., 2025) builds on LLaMA-2 (Touvron et al., 2023) to follow nuanced search instructions beyond semantic relevance. As LMs have demonstrated strong reasoning capabilities, we believe that high-quality, reasoning-aware retrieval representations are already latent in LM backbones (Hao et al., 2025); our goal is therefore to better elicit them and avoid overfitting during contrastive training, instead of re-learning narrow task-specific features from scratch.

**LM-guided Retriever Learning** While retrieval augments LM generation, LM can also improve retriever learning. Jiang et al. (2022) show that a single encoder-decoder transformer can jointly learn retrieval and reading end-to-

end, using attention patterns across questions and answers as supervision for retrieval. Similarly, Atlas (Izacard et al., 2023) utilizes cross-attention scores between retrieved documents and the generated output as signal to train the retriever. With the mainstream tendency of decoder-only LMs, RE-PLUG (Shi et al., 2024) improves retrieval by prepending retrieved documents to the queries and training retrievers by aligning retrievers' query-document similarity with the LM's perplexity. However, as the perplexity of frozen LMs is often poorly calibrated (Geng et al., 2024), this can lead to suboptimal retriever learning, and pairwise distillation introduces computational overhead. These issues are addressed in Revela (Cai et al., 2025), which connects the language modeling objective, i.e., NTP, to retriever learning via in-batch attention, enabling retrieval to be learned directly from raw text in a batch. Different from Revela, `ElicitR` uses the in-batch attention as a generative regularizer during supervised contrastive training on query–positive–negative batches, addressing contrastive overfitting and thereby improving reasoning-intensive retrieval.

**Reasoning-intensive Retrieval**   Unlike the classical retrieval datasets such as MS MARCO (Bajaj et al., 2016), which primarily evaluate keyword- or semantic-matching, reasoning-intensive retrieval, introduced by Su et al. (2025), requires deeper reasoning to identify relevant texts.

Such settings expose significant limitations of current off-the-shelf retrievers (Su et al., 2025). Most existing approaches generate synthetic reasoning-intensive query–document pairs and train retrievers contrastively on them (Shao et al., 2025; Das et al., 2025; Long et al., 2025). However, synthesis is complex, hard to validate, and difficult to scale. Only trained on raw text, Revela (Cai et al., 2025) achieves BRIGHT performance comparable to supervised retrievers and proprietary APIs, suggesting that learning directly from an LM can enhance retrievers' reasoning awareness beyond contrastive learning. Motivated by this observation, we take a different and orthogonal angle from data synthesis: recognizing that retriever backbones already encode substantial reasoning ability yet contrastive objectives often overfit, we ask how co-training with LMs can better regularize retriever learning so that this latent reasoning is more reflected in the learned representation.

**Distinction from Revela**   Although `ElicitR` and Revela (Cai et al., 2025) both leverage in-batch attention to couple language modeling with retrieval, their training regimes and the roles of this signal differ substantially. Revela is a *self-supervised pretraining* objective trained solely on raw text, with in-batch attention serving as the *sole* learning signal for the retriever. In contrast, `ElicitR` operates in a *supervised fine-tuning* setting and uses in-batch attention as an *auxiliary* generative regularizer alongside the InfoNCE

objective on labeled query-positive-negative batches. The two are therefore complementary rather than interchangeable: Revela learns retrieval-relevant structure from unlabeled corpora, whereas `ElicitR` exploits such structure to counteract contrastive overfitting during supervised retriever training.

## 3. Methodology: `ElicitR`

Our approach aims to enhance the reasoning capabilities of dense retrievers *without* relying on synthetic data generation. Instead, we seek to induce reasoning capacity in their intrinsic representations by mitigating overfitting during contrastive learning. We hypothesize the overfitting of contrastive learning comes from *over-simplification* of documents merely as positives and negatives given one query, which neglects the *rich* and *fine-grained* relationship among query and documents as general texts.

Beyond pure contrastive learning, we introduce generative regularization by injecting retriever-calculated similarities into LM training. Concretely, we train a *tiny* LM with NTP on the query together with its positive and negative documents. Taking query as example, generation is conditioned not only on their own prefix but also the corresponding positive and negative documents, with this conditioning modulated by the retriever. When two texts, whether they are queries or documents, are more closely related, the retriever increases their similarity, strengthening the LM's cross-conditioning signal. In this way, the retriever captures nuanced interactions among queries and documents, beyond binary query-document matching, and incorporates them as a generative regularizer that complements contrastive learning.

Following the motivation above, our algorithmic design is simple and practical. First, we initialize a retriever–LM framework that models relationships within text groups via in-batch generation on **only raw text**. Second, we use the same framework as a generative regularizer on **only query-document pairs**, alongside standard contrastive learning. Please refer to Appendix A for more algorithmic details.

### 3.1. Preliminaries

We employ a dual-encoder retriever $R_\theta$ that encodes a query $q$ and a document $d$ into dense vectors $R_\theta(q), R_\theta(d) \in \mathbb{R}^h$. Their similarity score is computed by a dot product, $s_\theta(q, d) = R_\theta(q)^\top R_\theta(d)$. We also utilize an LM $M_\phi$ which takes retrieved documents as context to predict tokens. The retriever and the LM are parameterized by $\theta$ and $\phi$, respectively. The goal of retriever learning is to assign higher similarity scores to more relevant query-document pairs, i.e., $s_\theta(q, d)$ should be larger when $d$ is more related to $q$.

### 3.2. Contrastive Learning with Generative Regularization

To align the retriever with fine-grained query–document relevance, we introduce a generative regularizer to mitigate the overfitting observed in contrastive learning. In a standard query-document setting, each query $q$ is paired with a positive document $d^+$ and a set of negatives $\{d_k^-\}$. For each query $q$, we form a batch $\mathcal{B} = \{q, d^+, d_1^-, \ldots, d_m^-\}$, and apply two losses on the batch.

#### 3.2.1. SUPERVISED CONTRASTIVE LOSS

We first optimize the retriever with a standard InfoNCE contrastive loss (Oord et al., 2018), encouraging higher scores for positives than negatives by minimizing

$$\mathcal{L}_{\text{InfoNCE}} = -\log \frac{\exp(s_\theta(q, d^+)/\tau_{\text{InfoNCE}})}{\sum_{d \in \mathcal{D}(q)} \exp(s_\theta(q, d)/\tau_{\text{InfoNCE}})}, \quad (1)$$

where $s_\theta(q, d)$ is the query-document similarity from $R_\theta$, $\mathcal{D}(q)$ is a joint set of positive and negative documents (including in-batch negatives), and $\tau_{\text{InfoNCE}}$ is a temperature.

#### 3.2.2. GENERATIVE REGULARIZATION

In addition to the contrastive loss, ElicitR co-trains a retriever and an LM via Revela-style generative regularization (Cai et al., 2025): the LM predicts each sequence while attending to the others, with attention weights $\alpha$ modulated by $R_\theta$. Specifically, for a batch $\mathcal{B} = \{x_1, \ldots, x_{|\mathcal{B}|}\}$, the weight $\alpha_{ij}$ that $x_i$ assigns to $x_j$ is defined as

$$\alpha_{ij} = \text{softmax}_{j \neq i}\left(\frac{s_\theta(x_i, x_j)}{\tau_{\text{genreg}}}\right), \quad (2)$$

where $s_\theta(x_i, x_j) = R_\theta(x_i)^\top R_\theta(x_j)$ is the similarity score computed by the retriever and $\tau_{\text{genreg}}$ is a temperature.

Concretely, for any $x \in \mathcal{B}$, the LM predicts its tokens conditioned on its own prefix and a retriever-weighted context over the remaining elements. For a sequence $x_i$, LM's standard causal self-attention is augmented with an *in-batch attention* over all other sequences $x_j$ ($i \neq j$) in the same batch, denoted as $x_{-i}$. The objective of generative regularization is the negative log-likelihood (NLL) of the next token in $x_i$, conditioned on both its local prefix $x_{i,<t}$ and a retriever-weighted in-batch context,

$$\mathcal{L}_{\text{genreg}} = -\sum_{i=1}^{|\mathcal{B}|} \sum_t \log P_{\theta,\phi}\big(x_{i,t} \mid x_{i,<t}, x_{-i}; \boldsymbol{\alpha}_i\big), \quad (3)$$

where the in-batch attention weights are $\boldsymbol{\alpha}_i = \{\alpha_{ij}\}_{j=1,j\neq i}^{|\mathcal{B}|}$, computed from retriever similarity scores produced by $R_\theta$.

We apply the generative regularizer uniformly to $q$, $d^+$, and negatives $\{d_k^-\}$, but for efficiency randomly sample fewer negatives than in InfoNCE.

**Layer-level operation.** ElicitR realizes in-batch attention as a parallel attention stream inside every transformer block of $M_\phi$, leaving the residual, LayerNorm, and FFN sublayers unchanged. At layer $l$, each sequence $x_i$ maintains two hidden states $[e_i^l; h_i^l] \in \mathbb{R}^{L \times d}$, where $e_i^l$ is the output of the standard causal self-attention (i.e., $M_\phi$'s original stream) and $h_i^l$ is the in-batch attention output. The standard self-attention stream is left unchanged,

$$\begin{aligned} (Q_i^e, K_i^e, V_i^e) &= e_i^{l-1}(W^Q, W^K, W^V), \\ e_i^l &= \text{softmax}\left(\frac{Q_i^e K_i^{e\top}}{\sqrt{d_H}}\right) V_i^e, \end{aligned} \quad (4)$$

with causal masking inside $x_i$ and head dimension $d_H$. The in-batch attention stream combines two components. First, a causal self-attention term computed on $h_i^{l-1}$ using the same projections,

$$s_i^l = \text{softmax}\left(\frac{Q_i^h K_i^{h\top}}{\sqrt{d_H}}\right) V_i^h. \quad (5)$$

Second, a cross-sequence term that queries the cached $(K_j^e, V_j^e)$ of every other sequence $x_j$ in the same batch under a full attention mask and aggregates them with the retriever-derived weights $\alpha_{ij}$ from Eq. 2,

$$b_i^l = \sum_{j=1, j\neq i}^{|\mathcal{B}|} \alpha_{ij} \, \text{softmax}\left(\frac{Q_i^h K_j^{e\top}}{\sqrt{d_H}}\right) V_j^e. \quad (6)$$

At every layer, the in-batch attention output is then formed by adding the cross-sequence contribution directly to the causal self-attention term, *before* the LayerNorm and FFN sublayers,

$$h_i^l = s_i^l + b_i^l, \quad (7)$$

so the cross-document signal is injected throughout the depth of $M_\phi$ rather than only at the output. Retriever gradients flow into $R_\theta$ through $\boldsymbol{\alpha}_i$ in Eq. 6. The cached keys and values $(K_j^e, V_j^e)$ in Eq. 6 are produced by a self-only forward pass on each $x_j$, so that every sequence's representation remains independent of the other in-batch sequences and is not contaminated by cross-document attention; see Appendix A.2 for the implementation.

**Overall Objective.** ElicitR's training objective combines the supervised contrastive signal with the generative regularization,

$$\mathcal{L}_{\text{ElicitR}} = \mathcal{L}_{\text{InfoNCE}} + \lambda \, \mathcal{L}_{\text{genreg}}, \quad (8)$$

where $\lambda$ controls the strength of the generative regularization. In this way, the contrastive loss enforces explicit query–document relevance, while ElicitR regulates the training.

Gradients from $\mathcal{L}_{\mathrm{genreg}}$ (i.e., NTP) update both $M_\phi$ and $R_\theta$, training $R_\theta$ to capture cross-document dependencies. In particular, the retriever is encouraged to assign *high* similarity scores $\alpha_{ij}$ to pairs $(x_i, x_j)$ where $x_j$ contains information *useful* for predicting tokens in $x_i$. This enables the retriever to effectively capture nuanced dependencies among queries and the associate documents within each batch, without incurring the quadratic pairwise comparison cost of Replug (Shi et al., 2024).

**Inference.** At test time, $M_\phi$ and the in-batch attention are discarded; ElicitR scores queries and documents with the dual-encoder alone, $s_\theta(q, d) = R_\theta(q)^\top R_\theta(d)$, adding no parameters, latency, or memory over a standard contrastive retriever.

### 3.2.3. GENERATIVE REGULARIZATION INITIALIZATION

LMs are typically trained to predict the next token conditioned only on the prefix of the same sequence, rather than on other sequences within the batch. To enable cross-sequence conditioning, we first initialize the joint retriever–LM framework, consisting of the retriever $R_\theta$ and LM $M_\phi$, on raw text (Cai et al., 2025). Concretely, we construct batches by selecting passages from a large unlabeled corpus (e.g., Wikipedia) and creating text chunks by splitting them, and jointly optimize both $R_\theta$ and $M_\phi$ with the same NTP objective (Eq. 3) on these raw-text batches (instead of query–positive–negative batches in Sec. 3.2.2).

## 4. Experiments

### 4.1. Experimental Setups

**Evaluation Benchmarks** To thoroughly evaluate our proposed framework, we assess ElicitR and the baselines on BRIGHT (Su et al., 2025), a standard, comprehensive, and diverse benchmark that requires intensive reasoning to retrieve relevant documents across a wide range of domains. To verify whether general capabilities are preserved, we further evaluate the models on BEIR (Thakur et al., 2021), a heterogeneous benchmark spanning multiple domains for general information retrieval.

**Training Data** We apply the query-document pairs for ElicitR, and use unlabeled raw texts without query-document pairs to initialize the generative regularizer. Specifically, we use MS MARCO,[1] a widely used large-scale dataset, as the *only* query-document supervision. It contains 491,007 pairs, where each query is corresponded to one positive documents and 15 negative ones in our training.

For the initialization of generative regularization, we follow

the original setups in Revela (Cai et al., 2025) to construct dataset from Wikipedia[2] and code-related corpus (Wang et al., 2025), including StackOverflow posts (Weber et al., 2024), online tutorials (Overwijk et al., 2022), and library documentations (Zhou et al., 2023). Wikipedia-only initialization yields comparable results (Sec. 5.1), suggesting that mixed-data initialization offers no domain advantage. We first segment passages into chunks of at most 120 words, ensuring that each chunk consists of complete sentences. Concretely, starting from passages $\{d_1, d_2, \ldots, d_n\}$, each passage $d_i$ is split into chunks $(d_{i1}, d_{i2}, \ldots, d_{im_i})$. We then arrange all chunks in a single sequence following document order, $(d_{11}, d_{12}, \ldots, d_{1m_1}, d_{21}, d_{22}, \ldots)$, and construct batches by taking consecutive segments from this sequence. In particular, batches are *allowed* to contain chunks drawn from several documents, which adds flexibility to the data construction in ElicitR's pre-training. In total, we collect and construct a pre-training corpus of 320,000 batches, each containing 16 chunks.

**Models** ElicitR involves the co-training between a retriever and a reference LM. Here, we take SmolLM2-135M (Allal et al., 2025) as the LM. For the retriever backbones, we consider a spectrum of LMs spanning 0.1B to 3B parameters to keep comparisons with prior work consistent: SmolLM2-135M, Qwen2.5-0.5B (Yang et al., 2024), LLaMA-3.2-1B (Grattafiori et al., 2024), and LLaMA-3.2-3B. We follow the recipe from Revela (Cai et al., 2025), where an `<eos>` token is appended to every chunk in the batch, and the embedding of this token is taken as the representation of the entire chunk. Moreover, we prepend the prefixes `query:` and `passage:` to the input queries and passages, respectively.

**Baselines** We compare our method against a pool of 12 retrieval systems that covers both sparse and dense paradigms and is broadly aligned with the set evaluated in BRIGHT (Su et al., 2025). As a classical sparse baseline, we use BM25 (Robertson et al., 2009), a strong lexical matcher highly competitive on BEIR (Thakur et al., 2021). For dense retrieval, we consider (i) compact open-source models, including Sentence-BERT (SBERT; Reimers & Gurevych, 2019) and BGE (Chen et al., 2024); (ii) large LM-based encoders with over 7B parameters, namely E5-Mistral-7B-Instruct (E5; Wang et al., 2024a), SFR-Embedding-Mistral (SFR; Meng et al.), and GritLM (Muennighoff et al., 2024); and (iii) explicitly instruction-aware models, Instructor-Large (Inst-L) and Instructor-XL (Inst-XL; Su et al., 2023). Many of these dense models are instruction-tuned and endowed with non-trivial reasoning ability. In addition, we evaluate several proprietary embedding mod-

---

[1] https://huggingface.co/datasets/Tevatron/msmarco-passage-aug

[2] https://huggingface.co/datasets/Tevatron/wikipedia-nq-corpus

els: `Cohere-embed-english-v3.0` (Cohere; [Cohere](#)), `voyage-large-2-instruct` (Voyage; [Voyage AI, 2024](#)), `text-embedding-3-large` (OpenAI; [OpenAI, 2024](#)), and `text-embedding-preview-0409` (Google; [Lee et al., 2024](#)).

Furthermore, we include RepLLaMA (Rep; [Ma et al., 2024](#)) at the same scales, trained identically to `ElicitR`'s contrastive component (including EOS pooling, prefixes, learning rate, temperature, etc) but optimizing only $\mathcal{L}_{\text{InfoNCE}}$ (i.e., without generative regularization).

**Experimental Details** For both stages, we use DeepSpeed ZeRO-3 and apply LoRA (rank 256, $\alpha=64$, dropout 0.1) to q_proj, k_proj, v_proj, o_proj, down_proj, up_proj, and gate_proj in the retriever and reference LM. Training is conducted in bf16 mixed precision.

**Generative Regularization Initialization** We jointly train the retriever and LM on the raw-text corpus described in Sec. 3.2.3. We train for 10,000 optimizer steps (one epoch) with a learning rate of $1\times10^{-4}$ using a linear scheduler with 100 warmup steps, a per-device batch size of 1, and gradient accumulation of 4 on 8 GPUs. Passages are truncated to 160 tokens, and we use EOS pooling with an appended `<eos>` token followed by $\ell_2$-normalization. The in-batch attention is controlled by an attention temperature $\tau = 1\times10^{-4}$.

**Generative Regularization with Contrastive Learning** We initialize both the retriever and the LM from their LoRA checkpoints above and train on MS MARCO. We keep the same LoRA configuration and contrastively train on 8 A100-SXM4-80GBs for one epoch (3,836 steps) with a learning rate of $5\times10^{-5}$, 100 warmup steps, a per-device batch size of 4 for contrastive learning, and gradient accumulation of 4. Queries and passages are truncated to 32 and 196 tokens, respectively. We set the contrastive temperature $\tau_{\text{InfoNCE}} = 0.01$ in Eq. 1 and the attention temperature $\tau_{\text{genreg}} = 1\times10^{-3}$ in Eq. 2, and weight the generative regularization by $\lambda = 1.0$ in Eq. 8. For the InfoNCE objective, we use 15 negatives per query. For efficiency, the generative regularizer is trained on a smaller subset, using batches that contain one query, its positive document, and 6 negatives (8 chunks in total). Again, we apply EOS pooling and $\ell_2$-normalization. Additional experimental details, including hyperparameter selection, are provided in Appendix B.1.

### 4.2. Experimental Results

**ElicitR greatly enhances retrievers' reasoning capacity without synthesizing query–document pairs.** As shown in Table 1, `ElicitR` consistently boosts reasoning-centric performance across nearly all datasets and model scales, while using *only* standard MS MARCO query–document pairs (and *no* synthetic reasoning-specific pairs). At 0.1B, `ElicitR` yields an average gain of 29%

over RepLLaMA, with particularly large improvements on reasoning-heavy StackExchange subsets such as EARTH. (+58%), ROB. (+106%), STACK. (+80%) as well as on the AoPS math benchmark (+64%). These gains remain consistent and strong at larger scales: at 0.5B and 1B scales, `ElicitR` improves over RepLLaMA by 16% and 21% on average, respectively. `ElicitR`-3B achieves the best overall nDCG@10 (23.1) among all open-source models reported in Table 1 on BRIGHT, surpassing even much larger 7B models fine-tuned on massive query-document pairs collected at scale or intensively prompted from LMs ([Wang et al., 2024a](#)), as well as proprietary API models. Moreover, it attains the highest number of task-wise first places, ranking top-1 on 7 tasks. This single-stage score even outperforms the strong **LLM reranking** baselines using GPT-4 reported in BRIGHT; since reranking depends on the candidate pool and inference budget, we use it only as a reference (in Table 10 in Appendix B.2). Together, these results indicate that generative regularization, co-training retrievers with a *small* LM (as small as 135M), is sufficient to endow dense retrievers with strong reasoning capabilities, without the cost or potential bias of synthetic supervision. We include a qualitative case study in Appendix B.3 where `ElicitR` retrieves the correct document despite far lower lexical overlap than RepLLaMA, illustrating how generative regularization helps move beyond surface-level matching.

**ElicitR matches synthetic-data baselines at a fraction of the resource cost.** Compared head-to-head with ReasonIR-8B ([Shao et al., 2025](#)), the state-of-the-art synthetic-data baseline on BRIGHT, `ElicitR`-3B trails by only 1.3 average nDCG@10 (23.1 vs. 24.4) while using $2.6\times$ fewer parameters, no synthetic supervision (491K MS MARCO pairs vs. 1.7M, of which roughly 345K are LM-synthesized), and no exposure to BRIGHT's corpus. The per-task breakdown (Table 11 in Appendix B.2) is complementary: `ElicitR`-3B wins on Bio., Earth., Econ., Rob., and Sus., while ReasonIR-8B retains its largest advantages on math and coding targeted tasks (Pony, AoPS, TheoQ., TheoT.). This supports our central claim that eliciting latent reasoning is a viable, resource-efficient direction that is *orthogonal* to data synthesis.

**ElicitR preserves the retrievers' capacity in general retrieval tasks.** We compare RepLLaMA and `ElicitR` on BEIR, including 13 subtasks, across four backbones from SmolLM2-135M to LLaMA-3.2-3B, as shown in Table 2. For all model sizes, `ElicitR` matches or slightly exceeds RepLLaMA in average nDCG@10, indicating no degradation in general-domain retrieval quality. Thus, `ElicitR` can be added to existing retrievers to boost performance on reasoning-intensive tasks (e.g., BRIGHT) while preserving effectiveness on broad-domain retrieval.

*Table 1.* **Retrieval performance (nDCG@10, %) across 12 tasks in BRIGHT.** ER is short for ElicitR. *Top-1 / 2* indicates the number of tasks where the model ranks 1st or 2nd. Small bottom-right percentages show ElicitR's relative gain over RepLLaMA at the same scale. **Bold** and underline indicate the best and second best performance across the models. Using MS MARCO as the *only* retrieval supervision and raw text for initialization of the generative regularizer, ElicitR outperforms the models with much more parameters and larger training datasets. Baselines except RepLLaMA are reported numbers from prior work (Su et al., 2025; Shao et al., 2025).

| Model | StackExchange | | | | | | | Coding | | Theorem-based | | | Avg. | Top-1/2 |
| | Bio. | Earth. | Econ. | Psy. | Rob. | Stack. | Sus. | Leet. | Pony | AoPS | TheoQ. | TheoT. | | |
|---|---|---|---|---|---|---|---|---|---|---|---|---|---|---|
| *Sparse model* | | | | | | | | | | | | | | |
| BM25 | 19.2 | 27.1 | 14.9 | 12.5 | 13.5 | 16.5 | 15.2 | 24.4 | 7.9 | 6.0 | 13.0 | 6.9 | 14.8 | 0 / 0 |
| *Open-sourced models (< 7B)* | | | | | | | | | | | | | | |
| SBERT | 15.1 | 20.4 | 16.6 | 22.7 | 8.2 | 11.0 | 15.3 | 26.4 | 7.0 | 5.3 | 20.0 | 10.8 | 14.9 | 0 / 0 |
| Rep-0.1B | 12.7 | 20.1 | 13.9 | 13.7 | 7.8 | 9.0 | 13.4 | 25.2 | 6.5 | 6.4 | 12.7 | 4.7 | 12.2 | 0 / 0 |
| ER-0.1B | 16.5₊₃₀% | 31.8₊₅₈% | 16.1₊₁₆% | 15.1₊₁₀% | 16.1₊₁₀₆% | 16.2₊₈₀% | 14.4₊₇% | 27.5₊₉% | 3.3₋₅₀% | 10.5₊₆₄% | 12.8₊₁% | 7.6₊₆₂% | 15.7₊₂₉% | 0 / 0 |
| Rep-0.5B | 14.4 | 28.3 | 17.1 | 18.9 | 17.8 | 15.4 | 12.7 | 27.4 | 3.2 | 12.0 | 21.0 | 8.6 | 16.4 | 0 / 0 |
| ER-0.5B | 20.7₊₄₄% | 37.2₊₃₁% | 20.3₊₁₉% | 20.6₊₉% | 19.8₊₁₁% | 17.6₊₁₅% | 16.9₊₃₃% | 28.5₊₄% | 6.1₊₈₈% | 12.8₊₆% | 19.3₋₈% | 8.5₋₁% | 19.0₊₁₆% | 0 / 2 |
| BGE | 11.7 | 24.6 | 16.6 | 17.5 | 11.7 | 10.8 | 13.3 | 26.7 | 5.7 | 6.0 | 13.0 | 6.9 | 13.7 | 0 / 0 |
| Inst-L | 15.2 | 21.2 | 14.7 | 22.3 | 11.4 | 13.3 | 13.5 | 19.5 | 1.3 | 8.1 | 20.9 | 9.1 | 14.2 | 0 / 0 |
| Rep-1B | 18.2 | 32.8 | 16.3 | 23.2 | 16.1 | 16.2 | 16.2 | 27.0 | 12.8 | 8.2 | 17.3 | 7.9 | 17.7 | 0 / 1 |
| ER-1B | 27.9₊₅₃% | 41.5₊₂₇% | 22.2₊₃₆% | 26.9₊₁₆% | 19.0₊₁₈% | 21.4₊₃₂% | 20.6₊₂₇% | 28.7₊₆% | 9.9₋₂₃% | 11.3₊₃₇% | 19.3₊₁₂% | 8.6₊₈% | 21.4₊₂₁% | 0 / 4 |
| Inst-XL | 21.6 | 34.3 | 22.4 | 27.4 | 18.2 | 21.2 | 19.1 | 27.5 | 5.0 | 8.5 | 15.6 | 5.9 | 18.9 | 0 / 1 |
| Rep-3B | 20.3 | 33.3 | 19.9 | **28.6** | 17.9 | 21.0 | 15.9 | 27.5 | 9.3 | **14.8** | 21.1 | 11.1 | 20.0 | 2 / 0 |
| ER-3B | **29.0**₊₄₃% | **44.1**₊₃₂% | 23.5₊₁₈% | 28.1₋₂% | **21.0**₊₁₇% | 23.4₊₁₁% | **22.4**₊₄₁% | **32.0**₊₁₆% | 3.0₋₆₈% | 12.8₋₁₃% | 24.8₊₁₈% | 12.6₊₁₃% | **23.1**₊₁₆% | 7 / 2 |
| *Open-sourced models (≥ 7B)* | | | | | | | | | | | | | | |
| E5 | 18.6 | 26.0 | 15.5 | 15.8 | 16.3 | 11.2 | 18.1 | 28.7 | 4.9 | 7.1 | **26.1** | **26.8** | 17.9 | 2 / 0 |
| SFR | 19.1 | 26.7 | 17.8 | 19.0 | 16.3 | 14.4 | 19.2 | 27.4 | 2.0 | 7.4 | 24.3 | 26.0 | 18.3 | 0 / 1 |
| GritLM | 25.0 | 32.8 | 19.0 | 19.9 | 17.3 | 11.6 | 18.0 | 29.8 | **22.0** | 8.8 | 25.1 | 21.1 | 20.9 | 1 / 1 |
| *Proprietary models* | | | | | | | | | | | | | | |
| Cohere | 18.7 | 28.4 | 20.4 | 21.6 | 16.3 | 18.3 | 17.6 | 26.8 | 1.9 | 6.3 | 15.7 | 7.2 | 16.6 | 0 / 0 |
| OpenAI | 23.7 | 26.3 | 20.0 | 27.5 | 12.9 | 12.5 | 20.3 | 23.6 | 2.5 | 8.5 | 23.8 | 12.3 | 17.8 | 0 / 0 |
| Voyage | 23.6 | 25.1 | 19.8 | 24.8 | 11.2 | 15.0 | 15.6 | 30.6 | 1.5 | 7.4 | **26.1** | 11.1 | 17.7 | 1 / 1 |
| Google | 23.0 | 34.4 | 19.5 | 27.9 | 16.0 | 17.9 | 17.3 | 29.6 | 3.6 | 9.3 | 21.5 | 14.3 | 19.5 | 0 / 0 |

*Table 2.* **Average BEIR performance by backbone (mean nDCG@10, %).** ElicitR matches RepLLaMA on BEIR while substantially improving BRIGHT. $\Delta$ is computed from unrounded values. Per-dataset results are in Table 9 in Appendix.

| Backbone | RepLLaMA | ElicitR | $\Delta$ |
|---|---|---|---|
| SmolLM2-135M | 43.3 | 43.7 | +0.4 |
| Qwen2.5-0.5B | 51.1 | 51.5 | +0.4 |
| LLaMA-3.2-1B | 53.6 | 53.8 | +0.2 |
| LLaMA-3.2-3B | 55.9 | 55.4 | -0.5 |

## 5. Analysis

### 5.1. Ablation on ElicitR

We ablate the training design of ElicitR using six variants: (i) **RepLLaMA**, the standard contrastive baseline; (ii) **ElicitR-*init***, the checkpoint obtained in Sec.3.2.3, which is trained on raw texts as the initialization of MS MARCO fine-tuning; (iii) **ElicitR w/o *init***, which directly co-trained the retriever and the LM on MS MARCO without initialization on raw texts; (iv) **ElicitR w. only *init* LM**, which initializes the retriever $R_\theta$ from the *base* backbone but uses the *initialized* LM $M_\phi$ from Sec. 3.2.3 for generative regularization during MS MARCO fine-tuning; (v) **ElicitR w. frozen LM**, which keeps that reference LM from the initialization fixed during generative regularization;

and (vi) **ElicitR**, our full method. The ablative results are presented in Table 3, and we conclude our findings as below.

- **Initialization of generative regularization alone is insufficient.** ElicitR-*init* performs poorly on BRIGHT (Avg. 13.8 at 1B and 15.8 at 3B), indicating that training in initialization alone with a *small* LM on raw text does not produce a strong reasoning-capable retriever compared to contrastive learning on query-document pairs.

- **However, initialization is critical for ElicitR.** Without it, ElicitR remains close to RepLLaMA at both scales (1B: 18.6 vs. 17.7; 3B: 20.3 vs. 20.0). Using only the initialized reference LM for generative regularization (ElicitR w. only *init* LM) improves performance to 19.3 (1B) and 20.9 (3B). Leveraging both the initialized retriever and LM (ElicitR) yields further gains, reaching 21.4 (1B) and 23.1 (3B). Overall, these results show that initialization is not a strong retriever by itself, but it *primes* ElicitR for effective generative regularization during MS MARCO fine-tuning.

- **Keeping LM trainable helps.** Freezing the reference LM (ElicitR w. frozen LM) performs slightly below the full method, trailing by 0.1 point at 1B (21.3 vs. 21.4) and 0.9 point at 3B (22.2 vs. 23.1). This gap suggests

*Table 3.* **Average BRIGHT performance across 12 tasks.** ElicitR consistently outperforms the contrastive baseline at every model scale. These gains stem from incorporating a generative regularizer initialized from *being trained* on raw texts, which can be *jointly* optimized with contrastive learning on the same data. $\Delta$ is computed from unrounded values. Please refer to Table 12 for per-task results.

| Method | 1B | 3B | $\Delta$ |
|---|---|---|---|
| RepLLaMA | 17.7 | 20.0 | 0.0 / 0.0 |
| ElicitR-*init* | 13.8 | 15.8 | -3.9 / -4.3 |
| ElicitR w/o *init* | 18.6 | 20.3 | +0.9 / +0.3 |
| ElicitR w. only *init* LM | 19.3 | 20.9 | +1.6 / +0.8 |
| ElicitR w. frozen LM | 21.3 | 22.2 | +3.7 / +2.1 |
| ElicitR | **21.4** | **23.1** | **+3.8 / +3.0** |

that allowing the reference LM to remain trainable better aligns the generative regularization with the distribution of contrastive training data.

Overall, ElicitR achieves the best performance at both scales, outperforming RepLLaMA and all ablated variants. These results show that initialization is necessary but not sufficient. The full training pipeline, including both initialization and generative regularization together with contrastive learning, is critical for reasoning-intensive retrieval.

**Where do the Pony and AoPS drops come from?** At 3B, ElicitR trails RepLLaMA on Pony (3.0 vs. 9.3) and AoPS (12.8 vs. 14.8), two tasks dominated by strict symbolic content (code syntax and math olympiad problems). Table 12 localizes the cause to the retriever initialization rather than the generative regularization itself: ElicitR w. only *init* LM, which keeps the retriever uninitialized and only uses the initialized reference LM, already recovers Pony to 11.1 and AoPS to 16.1, both above RepLLaMA. The drop therefore reflects a domain shift introduced by initializing the retriever on Wikipedia, which lacks formal symbolic text; a domain-specific initialization corpus is a natural remedy that we leave to future work.

**Wikipedia-only initialization.** To rule out domain-specific effects from code corpora, we rerun ElicitR with a *Wikipedia-only* initialization while keeping other training setups fixed. Performance remains comparable (1B: 21.9 vs. 21.4; 3B: 23.0 vs. 23.1), indicating the gains are not driven by the exposure of code-related corpora (Table 13). This further validates ElicitR's robustness to initialization.

### 5.2. ElicitR Prevents Overfitting

As shown in Figure 1a, directly fine-tuning an LM-based retriever on MS MARCO can lead to overfitting on reasoning-intensive benchmarks: after an early improvement phase, generalization performance may stagnate or degrade, sug-

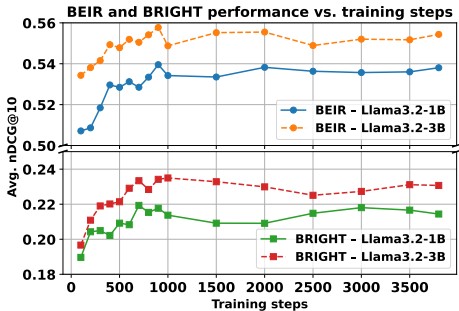

*Figure 2.* **Training dynamics during ElicitR (nDCG@10, %).** Average nDCG@10 on BEIR (top) and BRIGHT (bottom) across training checkpoints using LLaMA-3.2-1B/3B. Both curves improve and then plateau, showing that ElicitR mitigates MS MARCO fine-tuning overfitting (Figure 1a).

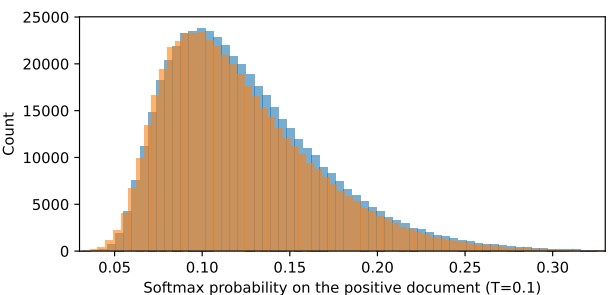

*Figure 3.* **Positive-document probability on MS MARCO (100-bin histogram; RepLLaMA vs. ElicitR).** ElicitR shifts the distribution toward lower confidence. $P(p^+ > 0.15)$: RepLLaMA 0.250 vs. ElicitR 0.228.

gesting that optimizing only on coarse query–document pairs is insufficient to preserve the LM's reasoning-oriented representations. ElicitR is motivated by mitigating this issue via a generative regularization objective, which encourages the model to capture finer-grained relationships between queries and documents, beyond isolated positive/negative pairs, and even between documents.

To validate this motivation, we evaluate intermediate checkpoints throughout fine-tuning on query-document pairs in ElicitR. Figure 2 shows that, both BEIR and BRIGHT performance improve early and then *remain stable* as training proceeds, without the characteristic late-stage degradation associated with overfitting. For per-task results, please refer to Appendix C.2.

### 5.3. Similarity Distribution Shift on MS MARCO

We analyze how ElicitR changes the similarity distribution during MS MARCO training. For each query, we compute a softmax over candidate scores $\{s(q,d)/T\}_{d \in \mathcal{C}}$ (1 positive + 15 negatives), where $T = 0.1$, and analyze the positive probability $p^+$. Figure 3 plots the empirical distribution of $p^+$ over $\sim$491K queries. Compared to contrastive-

*Table 4.* **Calibration on BRIGHT (Mean ECE, %, lower is better).** Confidence is derived from top-1 vs. top-2 score margin. $\Delta$ is computed from unrounded values. Please refer to Table 19 for per-task results.

| Backbone | RepLLaMA | ElicitR | $\Delta$ |
|---|---|---|---|
| SmolLM2-135M | 47.5 | **45.2** | $-2.3$ |
| Qwen2.5-0.5B | 48.0 | **45.4** | $-2.6$ |
| LLaMA-3.2-1B | 48.3 | **45.5** | $-2.7$ |
| LLaMA-3.2-3B | 45.9 | **43.8** | $-2.2$ |
| **Avg.** | **47.4** | **45.0** | $-2.4$ |

*Table 5.* **Average nDCG@10 (%) across three backbones and batch size** $\in \{4, 8, 12\}$**.** $\Delta$ is the max$-$min across batch sizes. For more details, please refer to Table 20 in Appendix C.5.

| Retriever Backbone | bz=4 | bz=8 | bz=12 | $\Delta$ |
|---|---|---|---|---|
| SmolLM2-135M | 15.2 | 15.7 | 15.6 | 0.5 |
| Qwen2.5-0.5B | 18.2 | 19.0 | 18.8 | 0.8 |
| LLaMA-3.2-1B | 21.3 | 21.4 | 21.9 | 0.6 |

only RepLLaMA, ElicitR produces a slightly less peaked distribution, shifting mass away from positive confidence, consistent with reduced contrastive overfitting.

**Alternative Entropy Regularization.** To test whether this effect alone explains the gains, we add an entropy regularizer to InfoNCE. It yields only small and inconsistent gains and remains clearly below ElicitR across backbones (Appendix C.3), suggesting that generic confidence smoothing alone is insufficient. ElicitR exploits nuanced signals among the query and its associated documents beyond the regularization constrained by binary relevance.

### 5.4. Calibration and Generalization on BRIGHT

In addition to retrieval quality, ElicitR yields a better-calibrated retriever compared with contrastive learning, i.e., that retrieval confidence from ElicitR can better reflect the probability of being correct. Concretely, for each query we estimate confidence from the score margin between the top-1 and top-2 retrieved documents as $p = \sigma(s_{(1)} - s_{(2)})$, where $\sigma$ is the sigmoid function (Svore et al., 2011). We define correctness by whether the top-1 result appears in the BRIGHT gold set. We then compute the Expected Calibration Error (ECE; Geng et al., 2024) by partitioning queries into 10 confidence bins and measuring, within each bin, the average gap between predicted confidence and empirical accuracy (lower ECE indicates better calibration).

Table 4 shows that ElicitR consistently **improves** calibration across backbones (135M-3B), reducing mean ECE relative to RepLLaMA. This trend complements our main effectiveness results: ElicitR not only improves nDCG@10 on reasoning-intensive retrieval, but also yields more *reliable* uncertainty. A natural explanation is that generative regularization discourages brittle, overly-confident separations produced by pure contrastive training. By requiring the retriever's similarity to modulate cross-attention among the query, positives, and negatives, the model learns more nuanced embeddings distribution, thereby reducing miscalibration. Please refer to Appendix C.4 for ECE details.

### 5.5. ElicitR's Practicality and Batch-size Robustness

While ElicitR effectively mitigates contrastive overfitting (Sec. 5.2) and substantially improves BRIGHT performance (Sec. 4.2), the in-batch NTP regularizer incurs additional compute and memory overhead compared to contrastive-only training. This overhead arises because NTP conditions each sequence not only on its own prefix but also on other texts within the same query-positive-negative batch, despite the small size of the LM. Consequently, smaller batches are more practical in terms of peak GPU memory.

Table 5 reports ElicitR performance on BRIGHT when varying the number of chunks for generative regularization to $k \in \{4, 8, 12\}$ (one query, one positive, and $k - 2$ negatives), across three model scales. Performance remains largely stable, indicating that ElicitR is robust to the query-positive-negative batch sizes. For the 3B retriever at $k = 4$, peak GPU memory rises from $\sim$17 GB (contrastive-only) to $\sim$27 GB with ElicitR (+10 GB / +59%). Despite this overhead, training remains feasible on 32–40 GB GPUs.

## 6. Conclusion

Real-world applications increasingly require stronger knowledge-seeking tools for complex scenarios, such as reasoning-intensive retrieval. Rather than *synthesizing* additional supervision, we investigate whether the reasoning capacity latent in LM-based retrievers can be *elicited* by mitigating overfitting via generative regularization. To this end, we propose ElicitR, a retriever-LM framework that models the nuanced relationships among queries and documents beyond binary relevance in contrastive learning. Given a batch containing a query and its candidate documents, NTP for each text is conditioned not only on its own prefix but also on other in-batch texts, with cross-text conditioning guided by retriever similarities. Experiments show that, using MS MARCO as the *only* paired query-document supervision and a small 135M LM for generative regularization, ElicitR consistently and substantially improves dense retriever performance on BRIGHT while maintaining general-domain capacity across backbones. Further analyses show that ElicitR improves reliability and remains robust even with small batch sizes, underscoring its practicality. In summary, we believe ElicitR offers a *fresh* perspective on learning stronger, more generalizable retrievers.

## Acknowledgements

We thank Sheng Lu, Tong Chen, Xinran Zhao, Sihao Chen, and Hongming Zhang for helpful discussions.

Fengyu Cai is funded by the Federal Ministry of Education and Research as part of the Software Campus 3.0 project ETRAG (funding code 16IS23067). This work is also supported by the Distr@l4a funding line of the State of Hesse (project number: 493 24 0015 4A). This work was also funded by the German Federal Ministry of Education and Research and the Hessian Ministry of Higher Education, Research, Science, and the Arts within their joint support of the National Research Center for Applied Cybersecurity ATHENE.

We gratefully acknowledge support from the hessian.AI Service Center (funded by the Federal Ministry of Research, Technology and Space, BMFTR, grant no. 16IS22091) and the hessian.AI Innovation Lab (funded by the Hessian Ministry for Digital Strategy and Innovation, grant no. SDIW04/0013/003).

## Impact Statement

Our work studies how to improve dense retrievers for reasoning-intensive search by reducing contrastive overfitting through co-training with a lightweight language model as a generative regularizer. This can benefit real-world information access by improving retrieval quality and reliability in domains where labeled query–document pairs are scarce, potentially lowering data-collection costs and enabling better downstream RAG and question answering. Potential risks include amplifying biases or misinformation present in the training corpora and increasing energy use if models are trained or deployed at scale; we mitigate these by using relatively small auxiliary LMs, reporting evaluation across diverse benchmarks, and encouraging dataset auditing and responsible deployment. We will release code and trained checkpoints to support reproducibility, facilitate independent bias/robustness testing, and help practitioners adopt the method with appropriate safeguards.

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

# A. Methodology

## A.1. Batch construction

For the initialization of generative regularization, we follow the configuration of Revela (Cai et al., 2025). The batch size is set 16 during initialization..

When using contrastive learning together with generative regularization, we use MS MARCO containing 491,007 query-document pairs. Here, we fill one batch with one query, its positive documents, and six corresponding negatives as shown in Figure 4. As analyzed in Sec.5.5, ElicitR is robust to small batch size, supporting its practicality.

---

1. Query: *average temperature in monterey ca*
2. Positive: *monterey bay, ca: Annual Weather Averages. August is the hottest month in monterey bay, ca with an average temperature of 18Â°C (64Â°F) and the coldest is January at 11Â°C (52Â°F) .onterey bay, ca: Annual Weather Averages. August is the hottest month in monterey bay, ca with an average temperature of 18Â°C (64Â°F) and the coldest is January at 11Â°C (52Â°F).*
3. Negative: *Most / Least Educated Cities in DE. The average temperature of Wilmington is 53.63Â°F, which is about the same as the Delaware average temperature of 54.48Â°F and is about the same as the national average temperature of 54.45Â°F.*
4. Negative: *The average annual precipitation / rainfall in Monterey is around 15 inches / 38 cm. The autumn season in Monterey is normally the warmest time of the year, with light rainfall and plenty of sunshine. Temperatures can reach around 21Â°C / 70Â°F.*
5. Negative: *California Weather ¿ Bodega Bay Weather Bodega Bay, CA Weather Bodega Bay, CA climate is mild during summer when temperatures tend to be in the 60's and cold during winter when temperatures tend to be in the 40's. The warmest month of the year is August with an average maximum temperature of 83.80 degrees Fahrenheit, while the coldest month of the year is December with an average minimum temperature of 34.30 degrees Fahrenheit.*
6. Negative: *This report describes the typical weather at the Monterey Peninsula Airport (Monterey, California, United States) weather station over the course of an average July. It is based on the historical records from 1974 to 2012. Earlier records are either unavailable or unreliable.*
7. Negative: *Burbank (Santa Clara County), CA Weather. Burbank (Santa Clara County), CA climate is warm during summer when temperatures tend to be in the 70's and cool during winter when temperatures tend to be in the 50's. The warmest month of the year is July with an average maximum temperature of 84.30 degrees Fahrenheit, while the coldest month of the year is December with an average minimum temperature of 41.00 degrees Fahrenheit.*
8. Negative: *Temperature. The warmest month in Metropolitan Oakland International, Oakland, California is September with an average high temperature of 73.2Â°F. The hottest day on record was September 28 2010 when the temperature hit 100.0Â°F.ain. The wettest month in Oakland, CA is March with an average of 1.6 inches of rain falling on 8 to 9 days. The driest month by average rainfall in Oakland, CA is August with an average of 0.1 inches of rain falling on 1 to 2 days.*

---

*Figure 4.* Example of one batch containing one query, one associated positive documents (green), and six negatives (blue).

## A.2. Retriever-weighted in-batch attention

We implement generative regularization jointly with contrastive learning by co-training a retriever $R_\theta$ and a lightweight LM $M_\phi$. Given a batch of sequences $\{D_i\}_{i=1}^B$ with tokens $D_i = \{x_1^i, \ldots, x_L^i\}$, the standard NTP objective is

$$\mathcal{L}_{\text{NTP}}(D_i) = -\sum_{l=1}^{L} \log P_\phi(x_l^i \mid x_{<l}^i). \tag{9}$$

In ElicitR, the NTP for each sequence is additionally conditioned on other sequences in the batch via retriever-weighted in-batch attention (Cai et al., 2025). Specifically, we compute similarity scores $s_{ij}$ using $R_\theta$ and convert them into attention weights $\alpha_{ij} = \text{softmax}_j(s_{ij}/\tau_{\text{genreg}})$, with $\sum_{j\neq i} \alpha_{ij} = 1$. The LM then attends to selected context sequences and the resulting NTP gradients provide an auxiliary learning signal to update $R_\theta$ through $\alpha_{ij}$.

*Table 6.* Backbone models used in `ElicitR` and their Hugging Face repositories and licenses.

| Model | Params | Hugging Face URL | License |
|---|---|---|---|
| SmolLM2-135M | 135M | https://huggingface.co/HuggingFaceTB/SmolLM2-135M | Apache-2.0 |
| Qwen2.5-0.5B | 0.5B | https://huggingface.co/Qwen/Qwen2.5-0.5B | Apache-2.0 |
| LLaMA-3.2-1B | 1B | https://huggingface.co/meta-llama/Llama-3.2-1B | LLaMA 3 Community License |
| LLaMA-3.2-3B | 3B | https://huggingface.co/meta-llama/Llama-3.2-3B | LLaMA 3 Community License |

*Table 7.* Overview of retrieval models evaluated in this work, which is collected from BRIGHT (Su et al., 2025).

| | Size | Max $|Q|$ | Max $|D|$ | Instruction | Version | License |
|---|---|---|---|---|---|---|
| | | | *Sparse model* | | | |
| BM25 | N/A | $\infty$ | $\infty$ | No | gensim | LGPL-2.1-only |
| | | | *Open-sourced models ($< 3B$)* | | | |
| SBERT | 109M | 512 | 512 | No | all-mpnet-base-v2 | Apache-2.0 |
| BGE | 335M | 512 | 512 | No | bge-large-en-v1.5 | MIT |
| Inst-L | 335M | 2048 | 2048 | Yes | instructor-large | Apache-2.0 |
| Inst-XL | 1.5B | 2048 | 2048 | Yes | instructor-xl | Apache-2.0 |
| | | | *Open-sourced models ($> 3B$)* | | | |
| E5 | 7.1B | 4096 | 4096 | Yes | e5-mistral-7b-instruct | MIT |
| GritLM | 7.1B | 256 | 2048 | Yes | GritLM-7B | Apache-2.0 |
| SFR | 7.1B | 4096 | 4096 | Yes | SFR-Embedding-Mistral | CC-BY-NC-4.0 |
| | | | *Proprietary models* | | | |
| Cohere | N/A | 512 | 512 | No | Cohere-embed-english-v3.0 | Company |
| Google | 1.2B | 2000 | 2000 | Yes | text-embedding-preview-0409, dim=768 | Company |
| OpenAI | N/A | 8191 | 8191 | No | text-embedding-3-large | Company |
| Voyage | N/A | 16000 | 16000 | Yes | voyage-large-2-instruct | Company |

**KV-cached in-batch attention.** In-batch attention is implemented by reusing cached key/value tensors. Concretely, we compute and cache $(K_i, V_i)$ for each context sequence $D_i$ (per layer) in a first pass. In a second pass, for a target sequence $D_i$, we compute $Q_i$ once and attend to the cached $(K_j, V_j)$ of selected neighbors. We then aggregate these cross-sequence attention outputs with similarity-based weights $\alpha_i$. For each sequence in each layer, we add the in-batch attention with self-attention in this second pass. Finally, at each layer we add this in-batch signal to the standard self-attention output, effectively separating the calculation of stardard-attention from cross-sequence context mixing.

## B. Experiments

### B.1. Setups

**Training Data** For the initialization of generative regularization, we sample the chunk batches from the batches constructed from Wikipedia and code-related corpus used in Revela (Cai et al., 2025). Each of them contains 160,000 batches, each containing 16 chunks. We included the examples in the supplementary materials.

**Models** Table 6 presents LM-based retriever backbones used in our experiments, together with their parameters, Hugging Face URLs, and the corresponding licenses.

**Baselines** Table 7 shows baseline models in the main experiment shown in Table 1 in Sec. 4.2.

**Hyperparameters** Table 8 presents how different temperature for in-batch attention weight, i.e., $\tau$genreg,[3] affect `ElicitR`'s performance. On one hand, across four model scales, `ElicitR` with the temperature $\tau = 10^{-3}$ generally outperform others. On the other hand, `ElicitR` consistently outperforms RepLLaMA, demonstrating its robustness to hyperparameters.

---

[3]For brevity, we use $\tau$ in the table.

*Table 8.* **Retrieval performance (nDCG@10, %) across 12 tasks in BRIGHT. Bold** and underline indicate the best and second best performance within each model-scale block. We sweep the in-batch attention weight $\tau$ used by `ElicitR` ($\tau_{genreg} = 10^{-k}$).

| Model | StackExchange | | | | | | | Coding | | Theorem-based | | | Avg. |
|---|---|---|---|---|---|---|---|---|---|---|---|---|---|
| | Bio. | Earth. | Econ. | Psy. | Rob. | Stack. | Sus. | Leet. | Pony | AoPS | TheoQ. | TheoT. | |
| *0.1B-scale models* | | | | | | | | | | | | | |
| RepLLaMA | 12.7 | 20.1 | 13.9 | 13.7 | 7.8 | 9.0 | 13.4 | 25.2 | **6.5** | 6.4 | 12.7 | 4.7 | 12.2 |
| ER ($\tau = 10^{-1}$) | 15.8 | 29.2 | 15.9 | 14.4 | 15.2 | 14.6 | **14.6** | 26.2 | 2.4 | 9.4 | **13.1** | **8.4** | 14.9 |
| ER ($\tau = 10^{-2}$) | 16.5 | 30.4 | 16.5 | **15.5** | 15.5 | 15.8 | 14.6 | 27.3 | 3.6 | 9.4 | 12.6 | 8.2 | 15.5 |
| ER ($\tau = 10^{-3}$) | 16.5 | 31.8 | 16.1 | 15.1 | **16.1** | **16.2** | 14.4 | 27.5 | 3.3 | 10.5 | 12.8 | 7.6 | **15.7** |
| ER ($\tau = 10^{-4}$) | **18.6** | **34.0** | **17.1** | 15.1 | 14.1 | 14.8 | 13.9 | 27.6 | 3.1 | **11.5** | 11.8 | 4.8 | 15.5 |
| *0.5B-scale models* | | | | | | | | | | | | | |
| RepLLaMA | 14.4 | 28.3 | 17.1 | 18.9 | 17.8 | 15.4 | 12.7 | 27.4 | 3.2 | 12.0 | **21.0** | 8.6 | 16.4 |
| ER ($\tau = 10^{-1}$) | 19.7 | **37.7** | 18.5 | 20.3 | 18.4 | 17.3 | 16.5 | 27.7 | **7.3** | 11.6 | 17.9 | **9.0** | 18.5 |
| ER ($\tau = 10^{-2}$) | 19.0 | 37.5 | 19.7 | 19.5 | **20.2** | 18.2 | 17.4 | 27.9 | 5.7 | 12.5 | 17.8 | 7.8 | 18.6 |
| ER ($\tau = 10^{-3}$) | 20.7 | 37.2 | **20.3** | 20.6 | 19.8 | 17.6 | **16.9** | **28.5** | 6.1 | **12.8** | 19.3 | 8.5 | **19.0** |
| ER ($\tau = 10^{-4}$) | **20.9** | 35.6 | 18.9 | **17.6** | 17.0 | 16.1 | 13.8 | 28.0 | 3.6 | 13.1 | 19.6 | 5.0 | 17.4 |
| *1B-scale models* | | | | | | | | | | | | | |
| RepLLaMA | 18.2 | 32.8 | 16.3 | 23.2 | 16.1 | 16.2 | 16.2 | 27.0 | **12.8** | 8.2 | 17.3 | 7.9 | 17.7 |
| ER ($\tau = 10^{-1}$) | 27.4 | **43.4** | 19.4 | 26.4 | 19.4 | 21.5 | 20.1 | 28.4 | 10.8 | 10.8 | 19.5 | **9.8** | 21.4 |
| ER ($\tau = 10^{-2}$) | 27.2 | 41.0 | **22.3** | 27.1 | **21.2** | 21.6 | 22.3 | 27.8 | 9.3 | 12.4 | **19.6** | 9.2 | **21.7** |
| ER ($\tau = 10^{-3}$) | **27.9** | 41.5 | 22.2 | **26.9** | 19.0 | 21.4 | 20.6 | 28.7 | 9.9 | 11.3 | 19.3 | 8.6 | 21.4 |
| ER ($\tau = 10^{-4}$) | 24.3 | 38.7 | 20.8 | 24.7 | 16.8 | 20.3 | 17.6 | **28.9** | 9.6 | **12.5** | 18.9 | 5.8 | 19.9 |
| *3B-scale models* | | | | | | | | | | | | | |
| RepLLaMA | 20.3 | 33.3 | 19.9 | 28.6 | 17.9 | 21.0 | 15.9 | 27.5 | **9.3** | **14.8** | 21.1 | 11.1 | 20.0 |
| ER ($\tau = 10^{-1}$) | **30.9** | **46.2** | 21.2 | 27.3 | 19.8 | 23.3 | 21.4 | 29.1 | 4.2 | 12.2 | 22.4 | 11.8 | 22.5 |
| ER ($\tau = 10^{-2}$) | 27.2 | 40.1 | 22.9 | 27.2 | **21.1** | 19.8 | 21.8 | 31.6 | 3.7 | 12.0 | 24.6 | 12.5 | 22.0 |
| ER ($\tau = 10^{-3}$) | 29.0 | 44.1 | 23.5 | 28.1 | **21.0** | 23.4 | **22.4** | **32.0** | 3.0 | 12.8 | **24.8** | **12.6** | **23.1** |
| ER ($\tau = 10^{-4}$) | 28.2 | 42.9 | **23.7** | **28.9** | 20.4 | 24.2 | **22.1** | 30.7 | 7.3 | 13.6 | 21.9 | 12.2 | 23.0 |

## B.2. Results

Table 9 reports per-task BEIR performance across four model scales. As detailed in Table 2, `ElicitR` maintains comparable general-domain retrieval performance to RepLLaMA, while substantially improving reasoning-intensive retrieval.

Table 10 reports the comparison between `ElicitR`, a single-stage retrieval, and LLM-based reranking algorithm, reported in BRIGHT (Su et al., 2025). They use sparse and dense retrievers to retrieve documents and rerank them by cross-encoder[4] and LLMs.[5] `ElicitR`'s performance at 3B scale outperforms these reranking baselines, demonstrating `ElicitR`'s effectiveness.

Table 11 reports a head-to-head per-task comparison between `ElicitR`-3B and ReasonIR-8B (Shao et al., 2025), the state-of-the-art synthetic-data baseline on BRIGHT, discussed in Sec.4.2. The two models match closely on average (23.1 vs. 24.4 nDCG@10) despite ReasonIR-8B using $2.6\times$ more parameters, $3.5\times$ more training pairs (with around 345K of them LM-synthesized), and synthesis seeded from BRIGHT's own corpus. The per-task pattern is complementary: `ElicitR`-3B wins on Bio., Earth., Econ., Rob., and Sus., where reasoning is largely about evidence aggregation over heterogeneous text, while ReasonIR-8B retains advantages on Pony, AoPS, TheoQ., and TheoT., where its math and coding targeted synthesis is most aligned with the task distribution.

## B.3. Qualitative Case Study: RSPO vs. Organic Certification

We trace one query from BRIGHT's `sustainable_living` subtask (qid 54) to illustrate the qualitative behavior of `ElicitR`.

The query reads: *"How environmentally friendly is palm oil with a certified organic label?"* RepLLaMA's top-1 document discusses RSPO certification, *"... Look for the RSPO label to ensure you purchase products made with certified sustainable palm oil..."*, which shares heavy surface overlap with the query ($\approx$62.5% word overlap on `certified`, `palm oil`, `environmentally responsible`) but answers about the *wrong* certification system.

---

[4] https://huggingface.co/cross-encoder/ms-marco-MiniLM-L12-v2
[5] The version of GPT-4 mentioned (Su et al., 2025) is `gpt-4-0125-preview`.

*Table 9.* Performance of retriever models on BEIR datasets (nDCG@10, %). **Bold** marks the best score per dataset.

| Model Size | 0.1B | | 0.5B | | 1B | | 3B | |
|---|---|---|---|---|---|---|---|---|
| **Dataset** | Rep | ElicitR | Rep | ElicitR | Rep | ElicitR | Rep | ElicitR |
| ArguAna | 44.7 | 46.5 | 51.1 | 50.6 | 50.5 | 49.7 | 52.4 | **53.2** |
| ClimateFEVER | 21.8 | 21.3 | 27.1 | 24.8 | 28.8 | 26.4 | **31.8** | 24.9 |
| DBPedia | 29.2 | 27.4 | 38.4 | 37.9 | 42.4 | 42.8 | **44.9** | 44.5 |
| FEVER | 61.9 | 65.9 | 77.0 | 79.9 | 79.8 | 81.5 | 82.8 | **83.2** |
| FiQA2018 | 26.9 | 29.5 | 36.7 | 37.5 | 40.3 | 41.2 | 43.1 | **44.0** |
| HotpotQA | 45.2 | 49.5 | 59.2 | 62.6 | 66.5 | 68.3 | 70.0 | **70.9** |
| NFCorpus | 31.7 | 29.8 | 33.8 | 33.5 | 35.7 | 36.4 | 37.2 | **37.8** |
| NQ | 41.8 | 40.8 | 52.6 | 52.6 | 57.8 | 58.9 | **62.0** | 62.0 |
| QuoraRetrieval | 84.5 | 85.1 | 83.6 | 85.6 | 84.4 | 85.8 | **86.5** | 84.4 |
| SCIDOCS | 11.8 | 13.4 | 15.5 | 16.4 | 17.8 | 18.2 | 18.7 | **19.2** |
| SciFact | 59.4 | 61.1 | 69.1 | 68.8 | 74.5 | 73.5 | 74.1 | **76.1** |
| TRECCOVID | 79.3 | 75.7 | 83.7 | 83.9 | 83.2 | 83.9 | 83.9 | **84.5** |
| Touche2020 | 24.0 | 22.6 | 36.1 | 34.9 | 35.4 | 32.8 | **39.1** | 35.8 |
| **Mean** | 43.3 | 43.7 | 51.1 | 51.5 | 53.6 | 53.8 | **55.9** | 55.4 |

*Table 10.* **Average reranking performance on BRIGHT (nDCG@10, %).** Reranking baselines are taken from BRIGHT (Su et al., 2025). We additionally report ElicitR-3B as a *single-stage* retriever for reference.

| Retriever | Reranker | $k$ | nDCG@10 |
|---|---|---|---|
| BM25 | None | – | 14.3 |
| | MiniLM | 10 | 13.1 |
| | MiniLM | 100 | 8.3 |
| | GPT-4 | 10 | 17.4 |
| | GPT-4 | 100 | 17.0 |
| Google | None | – | 19.5 |
| | MiniLM | 10 | 16.0 |
| | MiniLM | 100 | 9.4 |
| | GPT-4 | 10 | 21.5 |
| | GPT-4 | 100 | 22.6 |
| ElicitR-3B | None | – | **23.1** |

ElicitR, by contrast, retrieves a document that directly addresses organic certification, *"...organic labels do not establish standards against rainforest destruction and land grabbing..."*, despite a much lower surface overlap ($\approx$25.0%); identifying it requires the reasoning step organic $\neq$ sustainable even though their vocabularies overlap.

This example illustrates how contrastive-only training traps the retriever in surface-level semantic matching, while ElicitR's generative regularization encourages capturing distinctions beyond lexical overlap.

## C. Analysis

### C.1. Ablation Analysis

Table 12 reports per-task results for the ablated variants used in our analysis of ElicitR. Across BRIGHT subtasks, ElicitR consistently outperforms most of ablated baselines, underscoring the importance of its architecture and training pipeline.

Table 13 reports per-task results for the comparison between two different ElicitR's initialization. We find that ElicitR which is initialized with Wikipedia corpora will not degrade the performance, demonstrating that ElicitR is robust to the data used for initialization.

*Table 11.* **Head-to-head BRIGHT comparison (nDCG@10, %) between `ElicitR-3B` and ReasonIR-8B (Shao et al., 2025).** ReasonIR-8B uses 8B parameters and 1.7M training pairs, of which roughly 345K are LM-synthesized and part of the synthesis is generated from BRIGHT's own corpus. `ElicitR-3B` uses 3B parameters, 491K MS MARCO pairs only, and no BRIGHT-related data. **Bold** marks the better of the two per task. The two models match closely on average (1.3-point gap), with complementary per-task strengths.

| Model | StackExchange | | | | | | | Coding | | Theorem-based | | | Avg. |
|---|---|---|---|---|---|---|---|---|---|---|---|---|---|
| | Bio. | Earth. | Econ. | Psy. | Rob. | Stack. | Sus. | Leet. | Pony | AoPS | TheoQ. | TheoT. | |
| ReasonIR-8B | 26.2 | 31.4 | 23.3 | **30.0** | 18.0 | **23.9** | 20.5 | **35.0** | **10.5** | **14.7** | **31.9** | **27.2** | **24.4** |
| ER-3B | **29.0** | **44.1** | **23.5** | 28.1 | **21.0** | 23.4 | **22.4** | 32.0 | 3.0 | 12.8 | 24.8 | 12.6 | 23.1 |

*Table 12.* **Retrieval performance (nDCG@10, %) across 12 tasks in BRIGHT. Bold** and underline indicate the best and second best performance across the listed models. `ER` is the abbreviation of `ElicitR`.

| Model | StackExchange | | | | | | | Coding | | Theorem-based | | | Avg. |
|---|---|---|---|---|---|---|---|---|---|---|---|---|---|
| | Bio. | Earth. | Econ. | Psy. | Rob. | Stack. | Sus. | Leet. | Pony | AoPS | TheoQ. | TheoT. | |
| *1B-scale models* | | | | | | | | | | | | | |
| RepLLaMA | 18.2 | 32.8 | 16.3 | 23.2 | 16.1 | 16.2 | 16.2 | 27.0 | **12.8** | 8.2 | 17.3 | 7.9 | 17.7 |
| ER-*init* | 14.0 | 24.2 | 12.0 | 14.3 | 12.7 | 11.2 | 10.1 | 28.5 | 4.0 | 8.9 | 19.6 | 6.4 | 13.8 |
| ER w/o *init* | 21.0 | 36.6 | 18.8 | 25.1 | 15.4 | 16.0 | 15.5 | 27.8 | 11.6 | **12.6** | 16.7 | 6.0 | 18.6 |
| ER w. only *init* LM | 23.1 | 38.5 | 18.9 | 25.2 | 16.9 | 17.3 | 17.6 | 27.4 | 12.3 | 12.4 | 16.7 | 5.6 | 19.3 |
| ER w. frozen *init* LM | 25.6 | 41.0 | 21.0 | 25.9 | **19.5** | 19.9 | **20.6** | **29.1** | 11.4 | 11.3 | **20.5** | **10.1** | 21.3 |
| ER | **27.9** | **41.5** | **22.2** | **26.9** | 19.0 | **21.4** | 20.6 | 28.7 | 9.9 | 11.3 | 19.3 | 8.6 | **21.4** |
| *3B-scale models* | | | | | | | | | | | | | |
| RepLLaMA | 20.3 | 33.3 | 19.9 | **28.6** | 17.9 | 21.0 | 15.9 | 27.5 | 9.3 | 14.8 | 21.1 | 11.1 | 20.0 |
| ER-*init* | 14.1 | 28.8 | 14.4 | 15.3 | 13.5 | 15.2 | 15.1 | 29.8 | 2.4 | 9.6 | 22.0 | 9.3 | 15.8 |
| ER w/o *init* | 22.4 | 38.1 | 20.3 | 25.5 | 17.4 | 18.6 | 17.9 | 29.0 | 9.6 | 14.9 | 20.9 | 9.7 | 20.3 |
| ER w. only *init* LM | 23.9 | 38.3 | 21.2 | 27.4 | 17.8 | 19.3 | 17.7 | 29.1 | **11.1** | **16.1** | 19.7 | 9.0 | 20.9 |
| ER w. frozen *init* LM | 27.1 | 41.2 | 21.7 | 25.3 | 19.6 | 23.2 | 21.6 | **32.2** | 3.7 | 12.9 | 24.6 | **12.9** | 22.2 |
| ER | **29.0** | **44.1** | **23.5** | 28.1 | **21.0** | **23.4** | **22.4** | 32.0 | 3.0 | 12.8 | **24.8** | 12.6 | **23.1** |

## C.2. Training Dynamics

`ElicitR` is motivated by contrastive overfitting in retriever training. Tables 14 and 15 report per-task performance across RepLLaMA checkpoints during contrastive learning on MS MARCO, while Tables 16 and 17 report the corresponding results for `ElicitR`. As observed in Figure 1a and Figure 2 in Sec.5.2, `ElicitR` substantially mitigates this overfitting behavior.

## C.3. Alternative Entropy Regularization

Given in-batch similarities $s_{ij} = s(q_i, d_j)$, we form the distribution $p_{ij} = \mathrm{softmax}(s_{ij}/T_{\mathrm{entropy}})$ over candidates, i.e., one positive and 15 negative documents, for each query $q_i$. Entropy regularization encourages less peaked distributions by maximizing $H(\mathbf{p}_i) = -\sum_j p_{ij} \log p_{ij}$,

$$\mathcal{L} = \mathcal{L}_{\mathrm{InfoNCE}} - \lambda_{\mathrm{entropy}} \cdot \frac{1}{B} \sum_{i=1}^{B} H(\mathbf{p}_i),$$

where $\lambda_{\mathrm{entropy}} \geq 0$ controls the strength of smoothing.

Table 18 presents the results of using entropy regularization during contrastive learning on MS MARCO. Consistent with the main experiments, we include the results across four backbone scales, and sweep different setups of the temperature $T_{\mathrm{entropy}}$ and the strength $\lambda_{\mathrm{entropy}}$.

## C.4. Expected Calibration Error

In Sec.5.4, we evaluate whether a retriever's confidence reflects the probability that its top-1 retrieval is correct. For each query $q$, we define a scalar confidence using the margin between the top-1 and top-2 retrieval scores,

$$p(q) = \sigma\big(s_{(1)}(q) - s_{(2)}(q)\big), \tag{10}$$

*Table 13.* **Retrieval performance (nDCG@10, %) across 12 tasks in BRIGHT. Bold** and underline indicate the best and second best performance across the listed models. ElicitR is robust to the data used for initialization. This control isolates the effect of code-related corpora and indicates that our gains are not driven by exposure to code-related texts.

| Model | Bio. | Earth. | StackExchange Econ. | Psy. | Rob. | Stack. | Sus. | Coding Leet. | Pony | Theorem-based AoPS | TheoQ. | TheoT. | Avg. |
|---|---|---|---|---|---|---|---|---|---|---|---|---|---|
| | | | | | | *1B-scale models* | | | | | | | |
| RepLLaMA | 18.2 | 32.8 | 16.3 | 23.2 | 16.1 | 16.2 | 16.2 | 27.0 | **12.8** | 8.2 | 17.3 | 7.9 | 17.7 |
| ER | **27.9** | 41.5 | **22.2** | 26.9 | **19.0** | **21.4** | 20.6 | 28.7 | 9.9 | 11.3 | 19.3 | 8.6 | 21.4 |
| ER (Wiki-only init) | 26.3 | **42.6** | 21.8 | **28.4** | 17.8 | 20.4 | **22.0** | **29.2** | 10.3 | **12.7** | **20.4** | **10.4** | **21.9** |
| | | | | | | *3B-scale models* | | | | | | | |
| RepLLaMA | 20.3 | 33.3 | 19.9 | **28.6** | 17.9 | 21.0 | 15.9 | 27.5 | **9.3** | **14.8** | 21.1 | 11.1 | 20.0 |
| ER | 29.0 | **44.1** | 23.5 | 28.1 | **21.0** | 23.4 | **22.4** | **32.0** | 3.0 | 12.8 | **24.8** | **12.6** | **23.1** |
| ER (Wiki-only init) | **29.6** | 41.7 | **25.3** | 28.3 | 20.1 | **23.8** | 21.5 | 31.2 | 4.8 | 14.5 | 23.2 | 12.5 | 23.0 |

where $s_{(1)}(q)$ and $s_{(2)}(q)$ denote the scores of the highest- and second-highest-ranked documents for $q$, and $\sigma(\cdot)$ is the sigmoid function. We define correctness as

$$y(q) = \mathbf{1}\big[d_{(1)}(q) \in \mathcal{G}(q)\big], \tag{11}$$

where $d_{(1)}(q)$ is the top-1 retrieved document and $\mathcal{G}(q)$ is the gold set for $q$ in BRIGHT. To compute ECE, we partition queries into $B$ bins according to their confidence $p(q)$ (we use $B=10$). Let $\mathcal{Q}_b = \{q \mid p(q) \in I_b\}$ be the set of queries whose confidence falls into bin interval $I_b$. For each bin, we compute the empirical accuracy and average confidence:

$$\mathrm{acc}(\mathcal{Q}_b) = \frac{1}{|\mathcal{Q}_b|} \sum_{q \in \mathcal{Q}_b} y(q), \qquad \mathrm{conf}(\mathcal{Q}_b) = \frac{1}{|\mathcal{Q}_b|} \sum_{q \in \mathcal{Q}_b} p(q). \tag{12}$$

Finally, the Expected Calibration Error is the weighted average of the absolute gaps between accuracy and confidence:

$$\mathrm{ECE} = \sum_{b=1}^{B} \frac{|\mathcal{Q}_b|}{|\mathcal{Q}|} \, |\mathrm{acc}(\mathcal{Q}_b) - \mathrm{conf}(\mathcal{Q}_b)| \,. \tag{13}$$

Lower ECE indicates better calibration, i.e., predicted confidence more closely matches empirical correctness frequency.

As shown in Table 19, ElicitR's ECE is consistently lower than RepLLaMA, indicating that ElicitR is able to enhance the reliability of reasoning-intensive retrieval, beyond the retrieval quality.

### C.5. ElicitR's Batch Size Robustness

Table 20 presents ElicitR's per-task performance on BRIGHT with different query-positive-negative batch sizes across three different retriever scales. The performances on BRIGHT are consistent across batch sizes. This observation validate ElicitR's robustness to smaller batch sizes, further supporting its practicality.

*Table 14.* **Training dynamics on BEIR (per-dataset nDCG@10, %)** across MS MARCO fine-tuning checkpoints for Llama3.2-1B and Llama3.2-3B retrievers.

| Checkpoint | ArguAna | Climate | DBPedia | FEVER | FiQA | Hotpot | NF | NQ | Quora | SCIDOCS | SciFact | TREC | Touche | Mean |
|---|---|---|---|---|---|---|---|---|---|---|---|---|---|---|
| 1B-ckpt-100 | 46.5 | 24.8 | 36.1 | 75.7 | 36.8 | 59.7 | 33.8 | 48.2 | 87.0 | 17.9 | 69.0 | 75.6 | 24.7 | 48.9 |
| 1B-ckpt-200 | 50.1 | 21.8 | 36.1 | 72.5 | 38.1 | 62.4 | 34.4 | 51.2 | 87.5 | 17.6 | 71.7 | 76.2 | 20.5 | 49.2 |
| 1B-ckpt-300 | 48.7 | 25.4 | 38.8 | 81.2 | 38.7 | 63.2 | 35.7 | 54.7 | 87.5 | 17.4 | 69.7 | 82.2 | 27.0 | 51.6 |
| 1B-ckpt-400 | 50.5 | 28.0 | 39.5 | 80.0 | 39.3 | 64.2 | 36.1 | 55.3 | 87.8 | 17.9 | 73.6 | 81.4 | 27.3 | 52.4 |
| 1B-ckpt-500 | 49.9 | 28.6 | 38.6 | 82.4 | 38.4 | 63.9 | 36.1 | 55.0 | 87.0 | 17.4 | 73.0 | 83.9 | 28.8 | 52.5 |
| 1B-ckpt-600 | 50.6 | 27.3 | 38.0 | 81.8 | 38.9 | 63.5 | 35.9 | 53.7 | 87.5 | 17.9 | 73.2 | 83.6 | 30.4 | 52.5 |
| 1B-ckpt-700 | 52.0 | 31.8 | 40.2 | 81.0 | 39.7 | 64.4 | 36.0 | 55.9 | 87.9 | 18.3 | 73.0 | 83.2 | 32.6 | 53.5 |
| 1B-ckpt-800 | 51.0 | 27.6 | 40.1 | 80.7 | 40.2 | 64.4 | 35.8 | 55.1 | 86.9 | 17.4 | 74.6 | 82.2 | 35.3 | 53.2 |
| 1B-ckpt-900 | 49.5 | 30.9 | 40.2 | 80.9 | 39.7 | 65.3 | 35.9 | 55.8 | 86.4 | 17.7 | 74.7 | 83.4 | 34.0 | 53.4 |
| 1B-ckpt-1000 | 50.8 | 29.3 | 40.2 | 83.0 | 39.5 | 64.7 | 35.0 | 55.3 | 87.2 | 17.6 | 73.9 | 82.5 | 32.2 | 53.2 |
| 1B-ckpt-1500 | 49.7 | 31.0 | 41.6 | 82.9 | 39.4 | 65.8 | 36.6 | 56.8 | 86.3 | 18.6 | 76.6 | 83.7 | 33.7 | 54.1 |
| 1B-ckpt-2000 | 51.9 | 29.3 | 42.3 | 78.0 | 40.4 | 65.9 | 35.4 | 57.4 | 86.4 | 18.3 | 74.7 | 81.4 | 32.9 | 53.4 |
| 1B-ckpt-2500 | 51.5 | 29.5 | 43.3 | 80.1 | 39.5 | 67.0 | 36.0 | 57.4 | 85.0 | 18.0 | 74.6 | 81.3 | 34.7 | 53.7 |
| 1B-ckpt-3000 | 51.6 | 28.5 | 42.1 | 76.2 | 39.5 | 66.2 | 35.7 | 56.9 | 85.5 | 17.8 | 74.8 | 82.6 | 33.4 | 53.1 |
| 1B-ckpt-3500 | 51.0 | 28.6 | 42.5 | 78.7 | 39.4 | 66.6 | 35.4 | 57.1 | 84.4 | 17.5 | 74.2 | 81.9 | 34.8 | 53.2 |
| 1B-final | 50.5 | 28.8 | 42.4 | 79.8 | 40.3 | 66.5 | 35.7 | 57.8 | 84.4 | 17.8 | 74.5 | 83.2 | 35.4 | 53.6 |
| 3B-ckpt-100 | 49.9 | 30.1 | 39.2 | 83.0 | 40.8 | 63.9 | 35.6 | 54.9 | 87.8 | 20.4 | 73.4 | 81.2 | 28.4 | 53.0 |
| 3B-ckpt-200 | 52.9 | 30.8 | 40.2 | 82.7 | 42.8 | 64.9 | 36.1 | 56.4 | 88.2 | 19.8 | 74.7 | 83.9 | 27.8 | 53.9 |
| 3B-ckpt-300 | 49.1 | 32.4 | 41.1 | 84.0 | 43.1 | 65.1 | 36.7 | 58.4 | 88.0 | 19.9 | 73.6 | 85.5 | 31.4 | 54.5 |
| 3B-ckpt-400 | 52.1 | 32.1 | 41.7 | 83.2 | 42.2 | 66.6 | 37.0 | 59.1 | 88.3 | 19.2 | 74.8 | 84.7 | 31.7 | 54.8 |
| 3B-ckpt-500 | 53.3 | 32.0 | 42.1 | 82.1 | 42.9 | 68.3 | 37.6 | 59.8 | 87.6 | 20.0 | 76.3 | 85.0 | 34.0 | 55.5 |
| 3B-ckpt-600 | 54.8 | 32.9 | 42.5 | 85.2 | 42.9 | 67.5 | 37.0 | 59.2 | 87.8 | 20.2 | 74.1 | 84.1 | 36.0 | 55.7 |
| 3B-ckpt-700 | 53.5 | 33.6 | 41.5 | 82.5 | 41.9 | 67.2 | 37.0 | 58.8 | 87.4 | 19.5 | 74.9 | 83.3 | 33.7 | 55.0 |
| 3B-ckpt-800 | 52.6 | 31.4 | 42.4 | 82.0 | 42.6 | 68.7 | 37.3 | 59.7 | 87.4 | 19.2 | 75.9 | 83.5 | 40.2 | 55.6 |
| 3B-ckpt-900 | 51.5 | 33.2 | 42.7 | 83.5 | 43.0 | 68.8 | 36.3 | 60.1 | 87.4 | 19.1 | 75.1 | 84.8 | 38.9 | 55.7 |
| 3B-ckpt-1000 | 53.3 | 31.1 | 42.0 | 83.8 | 42.9 | 68.0 | 35.5 | 59.7 | 87.5 | 18.2 | 74.9 | 83.6 | 34.7 | 55.0 |
| 3B-ckpt-1500 | 53.0 | 32.8 | 43.1 | 84.9 | 42.0 | 68.7 | 36.4 | 59.1 | 86.8 | 18.9 | 75.6 | 84.2 | 39.5 | 55.8 |
| 3B-ckpt-2000 | 53.2 | 31.4 | 44.1 | 82.0 | 43.1 | 68.5 | 36.9 | 60.2 | 87.0 | 18.4 | 74.7 | 84.6 | 37.4 | 55.5 |
| 3B-ckpt-2500 | 53.0 | 31.3 | 44.3 | 80.7 | 42.1 | 69.1 | 36.7 | 59.9 | 86.8 | 18.0 | 75.6 | 83.5 | 36.5 | 55.2 |
| 3B-ckpt-3000 | 54.0 | 32.0 | 44.5 | 79.5 | 42.3 | 68.8 | 37.2 | 60.7 | 87.1 | 18.6 | 74.6 | 83.1 | 36.3 | 55.3 |
| 3B-ckpt-3500 | 52.9 | 32.6 | 44.4 | 82.5 | 42.2 | 69.8 | 36.5 | 61.1 | 86.5 | 18.2 | 74.5 | 82.7 | 39.2 | 55.6 |
| 3B-final | 52.4 | 31.8 | 44.9 | 82.8 | 43.1 | 70.0 | 37.2 | 62.0 | 86.5 | 18.7 | 74.1 | 83.9 | 39.1 | 55.9 |

*Table 15.* **Training dynamics on BRIGHT (per-task nDCG@10, %)** across MS MARCO fine-tuning checkpoints for Llama3.2-1B and Llama3.2-3B retrievers.

| Checkpoint | StackExchange | | | | | | | Coding | | Theorem-based | | | Avg. |
| | Bio. | Earth. | Econ. | Psy. | Rob. | Stack. | Sus. | Leet. | Pony | AoPS | TheoQ. | TheoT. | |
|---|---|---|---|---|---|---|---|---|---|---|---|---|---|
| 1B-ckpt-100 | 18.4 | 31.5 | 16.7 | 21.1 | 13.0 | 17.2 | 16.4 | 24.9 | 22.8 | 8.5 | 12.7 | 7.1 | 17.5 |
| 1B-ckpt-200 | 18.4 | 28.6 | 16.1 | 21.3 | 12.8 | 14.0 | 15.3 | 25.7 | 12.6 | 9.9 | 15.5 | 7.1 | 16.4 |
| 1B-ckpt-300 | 20.8 | 34.4 | 18.8 | 22.9 | 16.4 | 16.1 | 18.2 | 26.5 | 25.0 | 8.9 | 14.8 | 9.0 | 19.3 |
| 1B-ckpt-400 | 20.0 | 33.2 | 18.0 | 21.4 | 16.0 | 17.3 | 16.9 | 26.6 | 20.8 | 8.7 | 15.1 | 8.7 | 18.6 |
| 1B-ckpt-500 | 23.5 | 34.9 | 18.4 | 22.9 | 16.8 | 19.0 | 19.7 | 26.4 | 18.0 | 10.1 | 15.7 | 10.1 | 19.6 |
| 1B-ckpt-600 | 21.1 | 35.5 | 17.2 | 22.8 | 16.9 | 16.6 | 19.3 | 24.8 | 25.0 | 8.8 | 14.8 | 8.5 | 19.3 |
| 1B-ckpt-700 | 20.1 | 33.2 | 17.5 | 21.9 | 17.4 | 16.7 | 18.4 | 25.9 | 21.2 | 8.6 | 15.4 | 8.6 | 18.7 |
| 1B-ckpt-800 | 22.9 | 35.3 | 18.0 | 25.0 | 15.7 | 16.9 | 19.3 | 26.3 | 19.5 | 8.6 | 15.3 | 9.0 | 19.3 |
| 1B-ckpt-900 | 21.4 | 33.7 | 16.9 | 24.1 | 16.4 | 18.1 | 18.2 | 26.6 | 25.5 | 9.0 | 16.2 | 9.2 | 19.6 |
| 1B-ckpt-1000 | 20.6 | 33.3 | 17.7 | 22.5 | 15.9 | 17.8 | 17.1 | 25.7 | 21.3 | 8.2 | 14.0 | 9.1 | 18.6 |
| 1B-ckpt-1500 | 18.6 | 32.9 | 17.2 | 21.5 | 14.8 | 15.8 | 15.0 | 26.2 | 10.8 | 9.8 | 15.8 | 7.8 | 17.2 |
| 1B-ckpt-2000 | 20.1 | 35.1 | 17.2 | 27.3 | 14.7 | 17.1 | 15.4 | 26.1 | 19.0 | 8.9 | 17.6 | 9.1 | 19.0 |
| 1B-ckpt-2500 | 19.8 | 32.0 | 16.7 | 21.6 | 15.0 | 15.8 | 15.2 | 25.1 | 15.4 | 8.1 | 17.0 | 8.9 | 17.6 |
| 1B-ckpt-3000 | 19.7 | 33.2 | 16.7 | 22.6 | 15.7 | 16.1 | 15.9 | 26.2 | 12.1 | 8.0 | 17.3 | 8.3 | 17.7 |
| 1B-ckpt-3500 | 18.8 | 32.4 | 16.6 | 22.8 | 15.6 | 15.6 | 16.1 | 25.8 | 13.0 | 8.2 | 16.9 | 8.2 | 17.5 |
| 1B-final | 18.2 | 32.8 | 16.3 | 23.2 | 16.1 | 16.2 | 16.2 | 27.0 | 12.8 | 8.2 | 17.3 | 7.9 | 17.7 |
| 3B-ckpt-100 | 20.3 | 35.3 | 19.4 | 26.9 | 16.4 | 19.7 | 18.1 | 26.6 | 19.5 | 10.1 | 12.9 | 6.7 | 19.3 |
| 3B-ckpt-200 | 19.7 | 31.9 | 18.5 | 25.3 | 16.2 | 18.9 | 16.2 | 27.9 | 10.5 | 11.0 | 18.7 | 12.6 | 19.0 |
| 3B-ckpt-300 | 20.9 | 36.5 | 20.2 | 27.2 | 17.8 | 20.3 | 18.6 | 26.3 | 15.6 | 12.3 | 17.0 | 11.2 | 20.3 |
| 3B-ckpt-400 | 22.2 | 34.6 | 19.8 | 27.0 | 17.2 | 20.0 | 17.3 | 28.9 | 19.7 | 13.9 | 17.5 | 9.7 | 20.6 |
| 3B-ckpt-500 | 22.3 | 36.9 | 20.1 | 29.0 | 18.5 | 20.1 | 17.7 | 28.2 | 10.3 | 13.5 | 18.1 | 11.8 | 20.5 |
| 3B-ckpt-600 | 21.1 | 35.4 | 18.8 | 27.0 | 18.9 | 19.8 | 19.3 | 28.0 | 11.5 | 13.6 | 18.0 | 10.9 | 20.2 |
| 3B-ckpt-700 | 23.2 | 36.1 | 18.6 | 28.0 | 18.7 | 20.7 | 17.7 | 30.0 | 10.9 | 13.7 | 17.8 | 10.5 | 20.5 |
| 3B-ckpt-800 | 27.2 | 41.4 | 21.2 | 29.1 | 19.1 | 22.6 | 19.8 | 28.8 | 10.6 | 12.7 | 19.1 | 13.5 | 22.1 |
| 3B-ckpt-900 | 22.6 | 36.3 | 21.4 | 28.5 | 19.5 | 21.2 | 19.5 | 29.9 | 14.8 | 11.2 | 19.9 | 13.8 | 21.6 |
| 3B-ckpt-1000 | 21.1 | 35.8 | 19.8 | 26.9 | 18.4 | 22.8 | 17.1 | 28.7 | 11.5 | 12.3 | 18.9 | 10.4 | 20.3 |
| 3B-ckpt-1500 | 22.9 | 36.7 | 18.2 | 27.1 | 16.9 | 22.4 | 16.8 | 27.3 | 11.9 | 12.7 | 19.0 | 11.0 | 20.3 |
| 3B-ckpt-2000 | 22.3 | 36.0 | 21.2 | 28.5 | 18.8 | 20.5 | 17.6 | 26.9 | 12.5 | 15.3 | 19.7 | 9.1 | 20.7 |
| 3B-ckpt-2500 | 18.4 | 33.4 | 19.7 | 27.1 | 16.4 | 19.8 | 16.0 | 26.1 | 9.1 | 13.3 | 22.4 | 9.8 | 19.3 |
| 3B-ckpt-3000 | 19.7 | 34.3 | 19.6 | 28.0 | 17.2 | 20.3 | 16.0 | 26.9 | 6.5 | 13.5 | 21.6 | 10.6 | 19.5 |
| 3B-ckpt-3500 | 19.8 | 33.8 | 20.0 | 28.4 | 18.0 | 20.9 | 17.0 | 27.2 | 8.4 | 13.6 | 21.4 | 10.9 | 20.0 |
| 3B-final | 20.3 | 33.3 | 19.9 | 28.6 | 17.9 | 21.0 | 15.9 | 27.5 | 9.3 | 14.8 | 21.1 | 11.1 | 20.0 |

*Table 16.* **Training dynamics on BEIR (per-dataset nDCG@10, %)** across the checkpoints during `ElicitR` training for Llama3.2-1B and Llama3.2-3B retrievers.

| Checkpoint | ArguAna | Climate | DBPedia | FEVER | FiQA | Hotpot | NF | NQ | Quora | SCIDOCS | SciFact | TREC | Touche | Mean |
|---|---|---|---|---|---|---|---|---|---|---|---|---|---|---|
| 1B-ckpt-100 | 46.5 | 21.9 | 37.3 | 78.8 | 38.8 | 66.7 | 33.1 | 53.1 | 87.2 | 17.9 | 72.6 | 76.4 | 29.0 | 50.7 |
| 1B-ckpt-200 | 45.7 | 21.0 | 37.4 | 76.8 | 39.4 | 66.9 | 33.0 | 54.0 | 87.2 | 18.1 | 72.7 | 78.2 | 30.7 | 50.9 |
| 1B-ckpt-300 | 46.2 | 22.3 | 39.9 | 79.4 | 39.8 | 66.8 | 34.9 | 55.5 | 86.8 | 17.8 | 72.9 | 80.4 | 31.1 | 51.8 |
| 1B-ckpt-400 | 48.0 | 24.0 | 41.7 | 80.8 | 40.4 | 67.7 | 36.0 | 56.3 | 87.4 | 18.7 | 74.2 | 81.9 | 31.5 | 53.0 |
| 1B-ckpt-500 | 46.7 | 24.7 | 40.2 | 83.0 | 40.5 | 67.4 | 34.7 | 55.6 | 87.3 | 18.2 | 73.2 | 82.8 | 32.5 | 52.8 |
| 1B-ckpt-600 | 49.8 | 24.0 | 41.8 | 81.4 | 41.1 | 68.5 | 35.3 | 57.1 | 86.9 | 17.9 | 73.4 | 81.7 | 31.8 | 53.1 |
| 1B-ckpt-700 | 45.9 | 24.4 | 41.4 | 81.8 | 40.9 | 68.0 | 34.9 | 57.0 | 86.1 | 17.7 | 73.2 | 82.1 | 33.7 | 52.8 |
| 1B-ckpt-800 | 49.5 | 25.7 | 41.9 | 81.5 | 40.8 | 68.8 | 35.2 | 57.1 | 87.5 | 18.3 | 73.0 | 81.3 | 32.8 | 53.3 |
| 1B-ckpt-900 | 49.5 | 26.8 | 42.6 | 83.1 | 41.6 | 68.8 | 36.0 | 57.0 | 87.1 | 18.1 | 74.1 | 82.9 | 33.8 | 54.0 |
| 1B-ckpt-1000 | 49.8 | 25.7 | 41.4 | 82.5 | 40.0 | 68.3 | 35.2 | 57.3 | 86.8 | 17.6 | 73.7 | 82.6 | 33.7 | 53.4 |
| 1B-ckpt-1500 | 49.6 | 25.6 | 41.8 | 81.4 | 40.6 | 67.4 | 36.1 | 57.1 | 87.0 | 17.9 | 73.9 | 82.0 | 33.4 | 53.4 |
| 1B-ckpt-2000 | 50.9 | 26.6 | 42.8 | 80.3 | 41.7 | 68.2 | 36.3 | 57.9 | 86.9 | 18.6 | 74.5 | 82.6 | 32.4 | 53.8 |
| 1B-ckpt-2500 | 48.6 | 27.1 | 43.1 | 80.4 | 41.1 | 68.5 | 36.1 | 58.2 | 86.3 | 17.7 | 74.1 | 83.3 | 32.7 | 53.6 |
| 1B-ckpt-3000 | 49.0 | 25.6 | 43.0 | 80.6 | 41.2 | 68.6 | 36.1 | 58.3 | 86.0 | 17.7 | 73.9 | 83.2 | 33.3 | 53.6 |
| 1B-ckpt-3500 | 49.0 | 25.6 | 42.6 | 81.8 | 41.1 | 68.2 | 35.9 | 58.5 | 85.7 | 17.7 | 74.2 | 82.8 | 33.7 | 53.6 |
| 1B-final | 49.7 | 26.4 | 42.8 | 81.5 | 41.2 | 68.3 | 36.4 | 58.9 | 85.8 | 18.2 | 73.5 | 83.9 | 32.8 | 53.8 |
| 3B-ckpt-100 | 51.7 | 22.6 | 39.5 | 82.1 | 43.1 | 68.1 | 35.4 | 55.7 | 87.8 | 19.6 | 75.9 | 79.5 | 33.5 | 53.4 |
| 3B-ckpt-200 | 51.3 | 23.0 | 39.9 | 81.2 | 43.3 | 68.4 | 35.9 | 57.9 | 87.8 | 19.2 | 74.9 | 82.5 | 34.4 | 53.8 |
| 3B-ckpt-300 | 49.1 | 22.7 | 41.0 | 82.2 | 43.4 | 68.8 | 36.2 | 59.3 | 87.0 | 18.9 | 75.8 | 82.0 | 37.4 | 54.2 |
| 3B-ckpt-400 | 52.3 | 25.4 | 43.2 | 82.2 | 44.2 | 69.9 | 37.3 | 59.8 | 88.2 | 19.5 | 75.5 | 84.2 | 32.4 | 54.9 |
| 3B-ckpt-500 | 51.2 | 24.4 | 41.9 | 82.1 | 44.4 | 69.5 | 36.6 | 59.3 | 87.8 | 19.6 | 75.9 | 83.6 | 35.8 | 54.8 |
| 3B-ckpt-600 | 51.2 | 26.8 | 42.6 | 83.5 | 44.3 | 70.4 | 36.4 | 60.1 | 87.6 | 19.3 | 75.0 | 83.1 | 37.2 | 55.2 |
| 3B-ckpt-700 | 51.1 | 25.0 | 43.0 | 82.0 | 44.5 | 69.5 | 37.4 | 60.9 | 87.0 | 19.6 | 76.2 | 84.4 | 35.0 | 55.0 |
| 3B-ckpt-800 | 53.1 | 26.1 | 43.8 | 83.0 | 44.5 | 70.6 | 37.2 | 60.7 | 87.7 | 19.7 | 75.6 | 83.4 | 35.1 | 55.4 |
| 3B-ckpt-900 | 52.2 | 28.5 | 44.0 | 84.7 | 44.7 | 70.6 | 37.5 | 60.7 | 86.0 | 19.4 | 75.3 | 84.8 | 36.7 | 55.8 |
| 3B-ckpt-1000 | 51.0 | 24.3 | 43.4 | 82.7 | 44.3 | 70.4 | 37.1 | 60.9 | 85.4 | 19.1 | 75.3 | 83.5 | 35.9 | 54.9 |
| 3B-ckpt-1500 | 53.5 | 25.6 | 43.2 | 83.5 | 44.6 | 70.2 | 37.5 | 60.4 | 85.9 | 19.3 | 77.0 | 84.3 | 36.8 | 55.5 |
| 3B-ckpt-2000 | 53.9 | 25.5 | 44.4 | 83.1 | 44.9 | 70.5 | 38.0 | 61.8 | 85.3 | 19.3 | 76.4 | 83.8 | 35.2 | 55.6 |
| 3B-ckpt-2500 | 53.5 | 25.4 | 44.6 | 80.8 | 43.2 | 70.7 | 37.6 | 60.8 | 84.0 | 18.8 | 75.9 | 83.3 | 35.1 | 54.9 |
| 3B-ckpt-3000 | 54.1 | 24.5 | 44.9 | 81.8 | 44.1 | 71.0 | 37.4 | 61.9 | 84.0 | 19.0 | 75.7 | 84.1 | 35.1 | 55.2 |
| 3B-ckpt-3500 | 53.5 | 24.0 | 44.2 | 83.0 | 44.0 | 70.8 | 37.7 | 61.7 | 83.3 | 19.1 | 75.4 | 84.3 | 36.3 | 55.2 |
| 3B-final | 53.2 | 24.9 | 44.5 | 83.2 | 44.0 | 70.9 | 37.8 | 62.0 | 84.4 | 19.2 | 76.1 | 84.5 | 35.8 | 55.4 |

*Table 17.* **Training dynamics on BRIGHT (per-task nDCG@10, %)** across the checkpoints during `ElicitR` training for Llama3.2-1B and Llama3.2-3B retrievers.

| Checkpoint | StackExchange | | | | | | | Coding | | Theorem-based | | | Avg. |
|---|---|---|---|---|---|---|---|---|---|---|---|---|---|
| | Bio. | Earth. | Econ. | Psy. | Rob. | Stack. | Sus. | Leet. | Pony | AoPS | TheoQ. | TheoT. | |
| 1B-ckpt-100 | 24.6 | 38.7 | 16.6 | 21.6 | 17.5 | 17.6 | 18.1 | 28.1 | 5.1 | 11.2 | 20.2 | 8.4 | 19.0 |
| 1B-ckpt-200 | 27.6 | 41.6 | 18.8 | 22.7 | 18.4 | 20.0 | 21.3 | 28.1 | 6.6 | 12.1 | 19.9 | 8.0 | 20.4 |
| 1B-ckpt-300 | 27.2 | 40.8 | 19.1 | 23.0 | 18.4 | 20.4 | 20.9 | 27.9 | 8.2 | 11.9 | 20.1 | 8.0 | 20.5 |
| 1B-ckpt-400 | 25.0 | 40.2 | 19.5 | 23.4 | 18.4 | 19.7 | 17.9 | 29.3 | 8.4 | 12.4 | 19.8 | 8.5 | 20.2 |
| 1B-ckpt-500 | 26.9 | 43.5 | 19.9 | 24.0 | 18.6 | 20.6 | 19.0 | 29.0 | 8.9 | 12.7 | 19.7 | 8.3 | 20.9 |
| 1B-ckpt-600 | 26.4 | 41.9 | 19.5 | 24.1 | 18.4 | 19.2 | 19.7 | 28.3 | 10.4 | 12.4 | 20.9 | 8.8 | 20.8 |
| 1B-ckpt-700 | 29.4 | 44.2 | 20.7 | 25.6 | 19.1 | 20.9 | 21.6 | 28.6 | 10.9 | 13.3 | 20.2 | 8.7 | 21.9 |
| 1B-ckpt-800 | 28.1 | 42.8 | 19.6 | 26.5 | 19.0 | 20.3 | 19.5 | 29.5 | 11.9 | 12.8 | 20.5 | 8.1 | 21.5 |
| 1B-ckpt-900 | 27.2 | 42.7 | 22.0 | 26.9 | 18.9 | 20.8 | 19.3 | 29.6 | 12.0 | 12.2 | 20.5 | 9.0 | 21.8 |
| 1B-ckpt-1000 | 25.8 | 42.7 | 20.6 | 25.8 | 18.9 | 21.1 | 20.4 | 28.7 | 9.8 | 12.4 | 20.4 | 9.9 | 21.4 |
| 1B-ckpt-1500 | 27.6 | 40.6 | 20.0 | 26.2 | 19.0 | 21.0 | 19.1 | 28.0 | 10.5 | 13.1 | 18.7 | 7.3 | 20.9 |
| 1B-ckpt-2000 | 26.2 | 39.6 | 20.9 | 26.5 | 19.3 | 20.1 | 19.1 | 28.2 | 11.8 | 11.7 | 19.0 | 8.7 | 20.9 |
| 1B-ckpt-2500 | 28.0 | 41.9 | 20.1 | 28.0 | 18.7 | 20.4 | 20.7 | 28.9 | 12.4 | 11.5 | 18.4 | 8.9 | 21.5 |
| 1B-ckpt-3000 | 28.2 | 42.4 | 21.9 | 27.5 | 19.8 | 20.9 | 21.9 | 28.5 | 11.4 | 11.4 | 19.6 | 8.4 | 21.8 |
| 1B-ckpt-3500 | 28.6 | 42.9 | 21.6 | 26.6 | 18.9 | 21.4 | 21.8 | 27.6 | 11.4 | 11.9 | 19.2 | 8.2 | 21.7 |
| 1B-final | 27.9 | 41.5 | 22.2 | 26.9 | 19.0 | 21.4 | 20.6 | 28.7 | 9.9 | 11.3 | 19.3 | 8.6 | 21.4 |
| 3B-ckpt-100 | 23.5 | 36.9 | 18.3 | 21.8 | 17.4 | 20.7 | 19.4 | 31.6 | 2.9 | 11.2 | 22.4 | 9.9 | 19.7 |
| 3B-ckpt-200 | 28.2 | 41.2 | 19.0 | 23.0 | 19.1 | 22.4 | 21.2 | 32.4 | 3.5 | 11.8 | 21.6 | 9.4 | 21.1 |
| 3B-ckpt-300 | 29.3 | 45.4 | 20.0 | 25.3 | 18.7 | 22.9 | 23.0 | 32.1 | 4.4 | 12.2 | 20.6 | 9.0 | 21.9 |
| 3B-ckpt-400 | 27.0 | 43.8 | 20.5 | 26.1 | 18.3 | 22.4 | 22.2 | 32.3 | 6.1 | 12.2 | 22.3 | 11.1 | 22.0 |
| 3B-ckpt-500 | 28.6 | 43.7 | 20.7 | 25.5 | 19.1 | 22.3 | 22.5 | 32.0 | 5.1 | 13.1 | 22.2 | 10.9 | 22.1 |
| 3B-ckpt-600 | 32.1 | 46.4 | 21.5 | 27.0 | 20.2 | 22.0 | 23.5 | 32.2 | 4.6 | 13.0 | 22.2 | 10.3 | 22.9 |
| 3B-ckpt-700 | 31.2 | 44.8 | 22.9 | 27.3 | 21.2 | 23.8 | 24.7 | 32.8 | 3.5 | 13.2 | 22.5 | 12.1 | 23.3 |
| 3B-ckpt-800 | 30.8 | 45.7 | 21.3 | 26.5 | 19.8 | 23.2 | 23.2 | 32.3 | 3.9 | 13.1 | 23.1 | 11.4 | 22.8 |
| 3B-ckpt-900 | 31.6 | 45.8 | 23.1 | 27.1 | 20.9 | 23.4 | 23.4 | 33.2 | 4.3 | 13.1 | 23.3 | 11.9 | 23.4 |
| 3B-ckpt-1000 | 30.9 | 46.5 | 23.0 | 27.0 | 20.6 | 24.1 | 24.0 | 32.6 | 4.0 | 14.0 | 23.7 | 11.6 | 23.5 |
| 3B-ckpt-1500 | 29.9 | 46.8 | 22.7 | 27.7 | 21.2 | 24.5 | 22.3 | 31.2 | 5.1 | 12.9 | 23.5 | 11.6 | 23.3 |
| 3B-ckpt-2000 | 28.6 | 42.7 | 22.8 | 28.5 | 20.1 | 23.8 | 22.5 | 31.6 | 5.7 | 13.1 | 24.5 | 11.9 | 23.0 |
| 3B-ckpt-2500 | 28.9 | 42.7 | 22.6 | 27.2 | 20.4 | 22.3 | 22.9 | 31.4 | 4.0 | 12.0 | 23.8 | 11.9 | 22.5 |
| 3B-ckpt-3000 | 28.7 | 42.9 | 23.1 | 27.7 | 20.3 | 22.2 | 23.6 | 31.1 | 3.6 | 12.8 | 24.6 | 12.3 | 22.7 |
| 3B-ckpt-3500 | 28.8 | 44.3 | 23.0 | 28.4 | 21.0 | 22.9 | 24.1 | 31.4 | 3.3 | 13.3 | 24.3 | 12.6 | 23.1 |
| 3B-final | 29.0 | 44.1 | 23.5 | 28.1 | 21.0 | 23.4 | 22.4 | 32.0 | 3.0 | 12.8 | 24.8 | 12.6 | 23.1 |

*Table 18.* **Retrieval performance (nDCG@10, %) across 12 tasks in BRIGHT. Bold** and underline indicate the best and second best performance across the listed models. Training with entropy regularization will not improve the performance on BRIGHT. This control isolates distribution-flattening effects under InfoNCE training. Here, $T$ and $\lambda$ are short for $T_{\text{entropy}}$ and $\lambda_{\text{entropy}}$, respectively.

| | | | StackExchange | | | | | Coding | | Theorem-based | | | |
| Model | Bio. | Earth. | Econ. | Psy. | Rob. | Stack. | Sus. | Leet. | Pony | AoPS | TheoQ. | TheoT. | Avg. |
|---|---|---|---|---|---|---|---|---|---|---|---|---|---|
| | | | | | | *0.1B-scale models* | | | | | | | |
| RepLLaMA | 12.7 | 20.1 | 13.9 | 13.7 | 7.8 | 9.0 | 13.4 | 25.2 | 6.5 | 6.4 | 12.7 | 4.7 | 12.2 |
| Ent. ($T{=}0.02, \lambda{=}0.01$) | 11.8 | 17.3 | 13.9 | 12.7 | 6.7 | 7.8 | 14.0 | 25.6 | 5.8 | 5.8 | 12.7 | 5.2 | 11.6 |
| Ent. ($T{=}0.05, \lambda{=}0.01$) | 12.8 | 19.2 | 14.9 | 14.0 | 7.8 | 10.9s | 15.1 | 26.6 | 6.0 | 6.3 | 12.4 | 5.1 | 12.6 |
| ElicitR | 16.5 | 31.8 | 16.1 | 15.1 | 16.1 | 16.2 | 14.4 | 27.5 | 3.3 | 10.5 | 12.8 | 7.6 | 15.6 |
| | | | | | | *0.5B-scale models* | | | | | | | |
| RepLLaMA | 14.4 | 28.3 | 17.1 | 18.9 | 17.8 | 15.4 | 12.7 | 27.4 | 3.2 | 12.0 | 21.0 | 8.6 | 16.4 |
| Ent. ($T{=}0.02, \lambda{=}0.01$) | 15.5 | 30.0 | 17.6 | 19.7 | 19.7 | 15.4 | 12.4 | 28.0 | 2.7 | 10.3 | 20.3 | 9.1 | 16.7 |
| Ent. ($T{=}0.05, \lambda{=}0.01$) | 13.3 | 27.5 | 16.9 | 16.9 | 17.6 | 15.3 | 12.0 | 26.7 | 2.4 | 11.1 | 21.9 | 9.2 | 15.9 |
| ElicitR | 20.7 | 37.2 | 20.3 | 20.6 | 19.8 | 17.6 | 16.9 | 28.5 | 6.1 | 12.8 | 19.3 | 8.5 | 19.0 |
| | | | | | | *1B-scale models* | | | | | | | |
| RepLLaMA | 18.2 | 32.8 | 16.3 | 23.2 | 16.1 | 16.2 | 16.2 | 27.0 | 12.8 | 8.2 | 17.3 | 7.9 | 17.7 |
| Ent. ($T{=}0.02, \lambda{=}0.01$) | 19.0 | 34.9 | 16.3 | 22.0 | 15.8 | 16.1 | 16.4 | 26.7 | 13.3 | 8.9 | 17.2 | 7.5 | 17.8 |
| Ent. ($T{=}0.05, \lambda{=}0.01$) | 19.7 | 35.3 | 17.3 | 24.0 | 17.9 | 16.6 | 17.2 | 26.8 | **14.9** | 8.2 | 17.1 | 8.2 | 18.6 |
| Ent. ($T{=}0.02, \lambda{=}0.001$) | 17.2 | 33.4 | 16.3 | 22.7 | 16.4 | 15.9 | 16.5 | 25.9 | 14.0 | 8.3 | 17.0 | 7.8 | 17.6 |
| Ent. ($T{=}0.02, \lambda{=}0.0001$) | 19.3 | 34.8 | 16.9 | 22.5 | 16.4 | 16.2 | 16.1 | 26.4 | 12.1 | 8.4 | 17.0 | 8.7 | 17.9 |
| ElicitR | 27.9 | 41.5 | 22.2 | 26.9 | 19.0 | 21.4 | 20.6 | 28.7 | 9.9 | 11.3 | 19.3 | 8.6 | 21.4 |
| | | | | | | *3B-scale models* | | | | | | | |
| RepLLaMA | 20.3 | 33.3 | 19.9 | **28.6** | 17.9 | 21.0 | 15.9 | 27.5 | 9.3 | 14.8 | 21.1 | 11.1 | 20.0 |
| Ent. ($T = 0.02, \lambda = 0.01$) | 20.4 | 34.6 | 19.3 | 28.7 | 17.3 | 20.9 | 16.0 | 27.9 | 9.2 | 14.0 | 20.9 | 11.6 | 20.1 |
| Ent. ($T = 0.05, \lambda = 0.01$) | 21.8 | 35.6 | 20.1 | 27.9 | 17.4 | 21.8 | 17.0 | 27.1 | 6.8 | 13.2 | 21.1 | 11.9 | 20.1 |
| ElicitR | **29.0** | **44.1** | **23.5** | 28.1 | **21.0** | **23.4** | **22.4** | **32.0** | 3.0 | **12.8** | **24.8** | **12.6** | **23.1** |

*Table 19.* Calibration error (ECE, %) across 12 tasks in BRIGHT. Lower is better.

| | | | StackExchange | | | | | Coding | | Theorem-based | | | |
| Model | Bio. | Earth. | Econ. | Psy. | Rob. | Stack. | Sus. | Leet. | Pony | AoPS | TheoQ. | TheoT. | Avg. |
|---|---|---|---|---|---|---|---|---|---|---|---|---|---|
| | | | | | | *SmolLM2-135M* | | | | | | | |
| RepLLaMA | 44.9 | 41.7 | 51.9 | 46.3 | 51.0 | 48.7 | 48.5 | 38.3 | 50.5 | 48.2 | 44.2 | 55.6 | 47.5 |
| ElicitR | 46.3 | 26.9 | 46.3 | 49.3 | 44.8 | 45.9 | 48.6 | 37.0 | 54.9 | 45.0 | 44.1 | 53.3 | 45.2 |
| | | | | | | *Qwen2.5-0.5B* | | | | | | | |
| RepLLaMA | 51.5 | 36.6 | 48.6 | 45.9 | 49.2 | 46.7 | 51.6 | 42.4 | 58.6 | 46.1 | 42.3 | 56.5 | 48.0 |
| ElicitR | 42.3 | 23.4 | 46.2 | 46.3 | 43.1 | 50.9 | 50.3 | 42.5 | 51.6 | 50.2 | 45.2 | 52.7 | 45.4 |
| | | | | | | *Llama3.2-1B* | | | | | | | |
| RepLLaMA | 45.3 | 31.9 | 55.7 | 46.2 | 53.2 | 52.3 | 50.0 | 46.6 | 44.1 | 51.2 | 45.5 | 57.1 | 48.3 |
| ElicitR | 36.0 | 23.3 | 44.7 | 41.2 | 50.3 | 50.1 | 47.0 | 45.7 | 58.0 | 51.2 | 43.8 | 55.3 | 45.5 |
| | | | | | | *Llama3.2-3B* | | | | | | | |
| RepLLaMA | 43.3 | 34.1 | 47.6 | 42.8 | 51.9 | 51.0 | 47.7 | 45.4 | 49.2 | 42.7 | 40.0 | 55.5 | 45.9 |
| ElicitR | 33.4 | 19.1 | 40.2 | 40.9 | 53.5 | 44.8 | 44.7 | 41.1 | 63.0 | 49.0 | 40.8 | 54.7 | 43.8 |

*Table 20.* **Retrieval performance (nDCG@10, %) across 12 tasks in BRIGHT with different batch sizes (k=4, 8, 12). Bold** and underline indicate the best and second best performance across the listed models.

| | | | StackExchange | | | | | Coding | | Theorem-based | | | |
| Model | Bio. | Earth. | Econ. | Psy. | Rob. | Stack. | Sus. | Leet. | Pony | AoPS | TheoQ. | TheoT. | Avg. |
|---|---|---|---|---|---|---|---|---|---|---|---|---|---|
| | | | | | | *SmolLM2-135M* | | | | | | | |
| k=4 | 17.0 | 30.6 | 16.9 | 14.3 | 13.9 | 14.7 | 14.8 | 26.8 | 2.8 | 9.1 | 13.3 | 8.0 | 15.2 |
| k=8 | 16.5 | 31.8 | 16.1 | 15.1 | 16.1 | 16.2 | 14.4 | 27.5 | 3.3 | 10.5 | 12.8 | 7.6 | 15.6 |
| k=12 | 17.3 | 31.0 | 17.4 | 15.4 | 14.6 | 15.8 | 14.7 | 27.4 | 3.2 | 10.0 | 12.5 | 8.0 | 15.6 |
| | | | | | | *Qwen2.5-0.5B* | | | | | | | |
| k=4 | 19.2 | 35.4 | 19.1 | 20.9 | 19.5 | 17.3 | 16.5 | 27.2 | 6.2 | 11.9 | 18.5 | 7.3 | 18.2 |
| k=8 | 20.7 | 37.2 | 20.3 | 20.6 | 19.8 | 17.6 | 16.9 | 28.5 | 6.1 | **12.8** | 19.3 | 8.5 | 19.0 |
| k=12 | 20.7 | 37.5 | 19.8 | 20.2 | 19.6 | 17.9 | 17.1 | 28.6 | 4.3 | 12.6 | 19.3 | 7.9 | 18.8 |
| | | | | | | *LLaMA3.2-1B* | | | | | | | |
| k=4 | 25.9 | 41.2 | 20.6 | 26.3 | 19.9 | **21.9** | 20.3 | 28.3 | **10.3** | 11.9 | **20.0** | 9.2 | 21.3 |
| k=8 | 27.9 | 41.5 | 22.2 | 26.9 | 19.0 | 21.4 | 20.6 | 28.7 | 9.9 | 11.3 | 19.3 | 8.6 | 21.4 |
| k=12 | **28.0** | **42.6** | **22.8** | **27.7** | **20.0** | 21.4 | **22.0** | **28.8** | 9.4 | 11.7 | 19.7 | **9.4** | **21.9** |

