# OpenReview forum: "ElicitR: Unlocking Latent Reasoning in Dense Retrievers via Generative Regularization"
_ICML.cc/2026/Conference — ICML 2026 regular_

### Official Review · Reviewer_qhwS · 2026-03-09

**Soundness:** 3
**Presentation:** 3
**Significance:** 3
**Originality:** 3
**Overall Recommendation:** 5
**Confidence:** 1

**Summary:**

ElicitR is a retriever-language model co-training framework integrated with generative regularization. Building on contrastive learning, it jointly trains a small LM and realizes next token prediction through retriever-computed similarity-weighted cross-text conditioning. Using only MS MARCO as paired supervised data and unlabeled raw text for initialization, ElicitR achieves a relative performance improvement of 16-29% on the reasoning-intensive benchmark BRIGHT across retriever scales of 0.1B–3B (with an nDCG@10 of 23.1 at the 3B scale), while maintaining performance on the general retrieval benchmark BEIR. Additionally, it mitigates overfitting, enhances retrieval calibration, and remains robust to batch sizes.

**Compliance With Llm Reviewing Policy:**

Affirmed.

**Key Questions For Authors:**

1.Scalability of the Small Auxiliary Language Model: All experiments in the paper use the 135M-parameter SmolLM2 as the generative regularizer, even for retrievers with 3B parameters.
2.Exploration of the Root Cause of Contrastive Overfitting: The paper observes that fine-tuning on MS MARCO leads to a decline in reasoning capabilities. To what extent does this overfitting phenomenon depend on the quality of the MS MARCO dataset itself?
3.Portability of the Auxiliary Model: If the auxiliary model is portable, it will greatly enhance the flexibility of training new retrievers.

**Limitations:**

Yes

**Strengths And Weaknesses:**

Strengths:
1. The paper challenges the common perception that "synthetic data is indispensable for enhancing reasoning capabilities," demonstrating the value of the latent capabilities inherent in retrievers and reducing data construction costs.
2. The paper is the first to explicitly identify "contrastive overfitting" as the primary cause suppressing retrievers’ reasoning capacity, an insight that provides a new perspective for understanding the representation learning of dense retrievers.
3. It introduces retriever-computed similarities to dynamically control the intensity of LM’s context modulation, achieving deep alignment between the retrieval representation space and the generative semantic space.

Weaknesses:
1. The paper’s technical architecture heavily draws on Revela’s in-batch attention mechanism; its innovation lies more in how to apply this mechanism to address overfitting during supervised fine-tuning, rather than inventing a completely new underlying architecture.

---

> ### Author Rebuttal · Authors · 2026-03-30
>
> We thank the reviewer for the insightful evaluation and for recognizing ElicitR's novel perspective on contrastive overfitting, the value of eliciting latent reasoning capabilities, and the deep alignment between retrieval and generative semantic spaces. We believe these are indeed the core contributions of this work. We are happy to address the remaining questions below.
>
> ## W1: Technical Architecture Draws on Revela [1]
>
>
>
> While ElicitR adapts the in-batch attention mechanism from Revela and are
> transparent about this in the paper (Sec. 3.2.2), we hope to offer a **complementary**
> perspective: observing a novel phenomenon (the early-peak effect on BRIGHT), identifying
> its cause (contrastive overfitting suppresses latent reasoning), proposing a targeted
> solution (generative regularization), and validating its effectiveness (16–29% gains on
> BRIGHT across four backbones) together constitute substantial scientific contribution.
>
>
>
>
> To further illustrate the distinctions: Revela is a self-supervised pretraining framework without query-document pairs, while ElicitR targets supervised fine-tuning and addresses contrastive overfitting, entirely outside Revela's scope. In terms of functional role, Revela uses in-batch attention as its **sole** objective on **homogeneous** raw-text batches; ElicitR integrates it as an **auxiliary** regularizer on **query-positive-negative** batches alongside InfoNCE, and it is precisely this interaction that prevents overfitting. Table 3 confirms this empirically: ElicitR-init (Revela's setup) scores only 13.8/15.8 on BRIGHT, far below RepLLaMA (17.7/20.0), showing that the mechanism alone explains none of ElicitR's gains.
>
> ## Q1: Scalability of the 135M Auxiliary LM
>
> Thank you for this practical question. We note that the auxiliary LM serves as a **regularizer rather than a teacher model**:  it only needs sufficient capacity to make cross-text conditioning meaningful, not to be a strong language model in its own right. Empirically, ElicitR-3B achieves the strongest BRIGHT performance (23.1) among all open-source models including 7B+ models, and the gain over RepLLaMA remains **consistent from 0.1B (+3.5) to 3B (+3.1)**, suggesting no bottleneck at current retriever scales.
>
> Importantly, keeping the auxiliary LM small also has a **practical benefit**: as analyzed in Sec. 5.5, peak GPU memory at 3B increases by only **~10 GB (+59%)** over contrastive-only training, **remaining feasible on 32–40 GB GPUs**. Still, exploring larger auxiliary LMs remains an interesting direction, and we plan to investigate this in future work.
>
> ## Q2: Does Contrastive Overfitting Depend on MS MARCO Data Quality?
>
> This is an important question to disentangle. We believe the root cause is the **binary-label structure of contrastive learning rather than data quality**: InfoNCE reduces every document to a binary label regardless of the dataset, and as illustrated in Fig. 1(b), semantically related negatives and irrelevant ones receive identical near-zero supervision. This collapse is inherent to the contrastive objective and would occur with any dataset using binary relevance annotations. ElicitR therefore is motivated to model nuanced signals among queries, positive and negative documents.
>
> Two observations further support this. First, if data quality were the cause, BEIR would also degrade during training; instead, **only BRIGHT degrades while BEIR remains stable** (Fig. 1a). Second, ElicitR mitigates overfitting on the _same_ MS MARCO data simply by adding inter-text NTP conditioning (Fig. 2). If the data were the root cause, **regularization on the same data would not resolve the issue**.
>
> ## Q3: Portability of the Auxiliary Model
>
> We are glad the reviewer raised this, as portability is one of ElicitR's **practical strengths**. Table 3 demonstrates this directly: **ElicitR w. only init LM** pairs the initialized LM with a fresh retriever backbone and already achieves gains over RepLLaMA (1B: 19.3 vs. 17.7; 3B: 20.9 vs. 20.0), confirming that practitioners can **reuse our released initialized LM with any retriever backbone** of their choice without repeating the raw-text initialization. Joint initialization of both retriever and LM (full ElicitR) further improves to 21.4/23.1, as co-initialization better aligns the similarity space with the conditioning mechanism. We envision releasing **initialized retriever–LM pairs as reusable components**, requiring only ~10 GB additional GPU memory at 3B scale (Sec. 5.5).
>
> We hope these clarifications further strengthen the reviewer's confidence in ElicitR's contributions, and are happy to discuss further if needed.
>
> [1] Revela: Dense Retriever Learning via Language Modeling, ICLR 2026

---

> > ### Author Rebuttal · Reviewer_qhwS · 2026-04-06
> >
> > I have read the opinions of the other reviewers, and the author's rebuttal has addressed my concerns. I therefore maintain my Accept recommendation.

---

> > > ### Author Response · Authors · 2026-04-07
> > >
> > > Dear Reviewer qhwS,
> > >
> > > Thank you for the careful reading and for confirming that your concerns have been fully addressed. We are glad the rebuttal was helpful. Given your full resolution, we would kindly invite you to also consider updating your confidence score if you feel more certain about the assessment after reading the rebuttal and the discussion among reviewers.
> > >
> > > We remain happy to discuss further if needed.
> > >
> > > Best,
> > >
> > > Authors of ElicitR

---

### Official Review · Reviewer_oQHP · 2026-03-12

**Soundness:** 3
**Presentation:** 4
**Significance:** 3
**Originality:** 3
**Overall Recommendation:** 4
**Confidence:** 3

**Summary:**

This paper proposes ElicitR, a retriever-LM co-training framework that addresses contrastive overfitting (a key barrier to reasoning in dense retrievers) via generative regularization—augmenting contrastive learning with a small 135M LM for cross-text next token prediction (weighted by retriever similarities). Trained only on MS MARCO and unlabeled text (no costly synthetic reasoning pairs), ElicitR delivers 16–29% relative gains on reasoning-intensive BRIGHT (3B model hits 23.1 nDCG@10, outperforming 7B models/proprietary APIs) while preserving BEIR general-domain performance. It mitigates overfitting, improves calibration, and is robust to small batches, validating practicality.

**Compliance With Llm Reviewing Policy:**

Affirmed.

**Key Questions For Authors:**

See weakness

**Strengths And Weaknesses:**

Strength:

* The paper identifies an underexplored issue in dense retrieval: LM-based retrievers already encode latent reasoning ability, which is suppressed by contrastive overfitting to binary relevance.
* The method uses only MS MARCO data, requires no synthetic supervision, and is compatible with existing retriever backbones, making it practical for adoption.
* Experiments are thorough and cover multiple dimensions: cross-scale validation (0.1B–3B models), strong performance on BRIGHT, maintained generalization on BEIR, and detailed ablations validating key design choices.
* The paper is clearly written with intuitive figures explaining the motivation and mechanism.

Weakness:

* ElicitR performs poorly on the AoPS benchmark (math reasoning), with large drops compared to RepLLaMA. The paper does not analyze this failure mode, leaving uncertainty about the method’s ability to handle formal reasoning tasks.
* While outperforming standard baselines, the paper provides limited comparison with recent synthetic reasoning data approaches (e.g., ReasonIR, RaDeR). A more direct comparison would better validate the proposed paradigm.
* The paper shows that generative regularization helps but does not analyze what semantic signal the NTP objective contributes to retriever learning (e.g., reasoning vs. lexical dependencies). Ablations isolating retriever-weighted attention would improve understanding.

---

> ### Author Rebuttal · Authors · 2026-03-30
>
> We thank the reviewer for the thorough evaluation and for recognizing ElicitR's practical value, the underexplored nature of contrastive overfitting, and the thoroughness of our experiments. We would address each concern below.
> ## W1: AoPS (Math Reasoning) Performance
>
>
> We appreciate the reviewer highlighting the performance degradation on AoPS. Our ablation study (**Sec.5.1 and Table 11**) reveals this drop is caused by the usage of general corpus (e.g., Wikipedia) in the **initialization phase**, not the generative regularization mechanism itself.
>
>
> The key evidence is the **ER w. only init LM** setting, which pairs an initialized auxiliary LM with an _uninitialized_ retriever backbone. This isolates the effect of the regularization from the retriever initialization. Under this setting, AoPS recovers to 16.1, **surpassing** RepLLaMA's 14.8. This directly demonstrates that the regularization mechanism itself benefits formal reasoning; the degradation stems solely from retriever initialization.
>
>
> In short, the mechanism is **not** the culprit. The root cause is a data-centric domain shift during initialization: the generic corpus  lacks formal symbolic reasoning data, shifting the retriever's latent space away from these domains (ER-init scores only 9.6 on AoPS). Across most of the remaining tasks, ElicitR generally and substantially outperforms RepLLaMA.
>
> Accordingly, we plan to curate a broader, domain-specific initialization corpus to fully realize ElicitR's potential on these highly specialized tasks.
>
>
> ## W2: Comparison with Synthetic Reasoning Data Approaches
> **ElicitR-3B closely matches ReasonIR-8B despite fundamentally different resource requirements.** ReasonIR-8B [1] achieves 24.4 nDCG@10 on BRIGHT. ElicitR-3B reaches 23.1 with **2.6× fewer** parameters, and **no synthetic data** (491K MS MARCO pairs vs. 1.7M including ~345K synthesized). Moreover, ReasonIR synthesizes **part of training data from BRIGHT's corpus**, giving it a direct domain advantage. ElicitR uses no BRIGHT-related data, making the 1.3-point gap notable.
>
>
> The per-task comparison reveals complementary strengths:
> | | Bio. | Earth. | Econ. | Psy. | Rob. | Stack. | Sus. | Leet. | Pony | AoPS | TheoQ. | TheoT. | **Avg.** |
> |---|---|---|---|---|---|---|---|---|---|---|---|---|---|
> | ReasonIR-8B | 26.2 | 31.4 | 23.3 | **30.0** | 18.0 | **23.9** | 20.5 | **35.0** | **10.5** | **14.7** | **31.9** | **27.2** | **24.4** |
> | ElicitR-3B | **29.0** | **44.1** | **23.5** | 28.1 | **21.0** | 23.4 | **22.4** | 32.0 | 3.0 | 12.8 | 24.8 | 12.6 | 23.1 |
>
>
> We believe this comparison validates our central claim: eliciting latent reasoning through generative regularization is **viable and resource-efficient** as a **novel and complementary** direction to data synthesis that addresses contrastive overfitting with lower complexity.
>
>
> ## W3: What Semantic Signal Does NTP Contribute?
> Thank you for raising this question. We kindly note that **Table 3** already provides ablations that directly isolate the retriever-weighted attention signal, and the results collectively show that the NTP signal is **content-level and reasoning-oriented, not lexical**.
>
>
> **The signal encodes nuanced relevance beyond binary labels.** The retriever must produce $\alpha$ weights such that cross-text conditioning helps token prediction: for the query (qid `45292` in MS MARCO) in Fig 1(b), NegX (topically related) receives 0.48 while NegY (irrelevant) receives 0.00, while InfoNCE collapses to identical near-zero.
>
>
> **The signal is reasoning-level, not lexical.** If NTP captured lexical dependencies, BEIR gains would be proportional to BRIGHT gains. The opposite holds: 16-29% relative improvement on BRIGHT vs. ±0.5 on BEIR (Table 2), confirming that NTP specifically captures the deeper cross-text relationships that reasoning requires.
>
>
> **Existing ablations isolate the retriever-weighted attention.** Table 3 provides three controls that together precisely characterize the NTP signal:
> - **ElicitR-init** is the most direct isolation: it trains _only_ with retriever-weighted NTP on raw text (no contrastive learning at all). It scores 13.8/15.8, below RepLLaMA (17.7/20.0), showing that NTP alone is insufficient. Instead, it is a **complementary** regularizer that interacts with InfoNCE to unlock reasoning.
> - **Full ElicitR** combines both objectives and reaches 21.4/23.1, where the gain over RepLLaMA (+3.7/+3.1) quantifies the net NTP contribution as a regularizer, while the gap to ElicitR-init (+7.6/+7.3) confirms the **synergy** between contrastive method and generative regularization.
> - **Frozen LM** isolates the gradient pathway: when the LM is frozen, the _only_ path from NTP to $R_\theta$ is through $\alpha$. It still achieves +3.6/+2.2 over RepLLaMA, proving the **contribution** of the retriever-weighted attention itself.
>
> We believe these results collectively address the concern and are happy to discuss further.
>
> [1] ReasonIR: Training Retrievers for Reasoning Tasks, COLM 2025

---

> > ### Author Rebuttal · Reviewer_oQHP · 2026-04-01
> >
> > Thanks for the authors’ rebuttal. I have read it carefully, and I acknowledge that it partially addresses my concerns by providing additional empirical results. I also agree that this is an interesting topic. As noted by the other reviewers, the observed limitation does not seem to hurt formal reasoning tasks, but rather appears to stem from the retriever’s initialization phase.
> >
> > That said, the rebuttal still does not clearly explain how the proposed method would perform when the retriever is already strong, nor does it fully address its limited robustness under data-centric domain shifts during initialization. In other words, the method’s effectiveness across different settings and its boundaries of applicability remain insufficiently clarified.
> >
> > Overall, I find the work inspiring, and I encourage the authors to continue exploring this problem, especially by better characterizing the broader range of scenarios where the method is effective, as well as the situations in which it may not be suitable. For these reasons, I will keep my score unchanged.

---

> > > ### Author Response · Authors · 2026-04-01
> > >
> > > Dear Reviewer oQHP,
> > >
> > > Thank you for the careful reading and for maintaining a positive assessment of the work.
> > >
> > > We appreciate your acknowledgement that the observed limitation **does not hurt formal reasoning tasks**, which largely addresses the core concern in the original review. The two remaining points, i.e., performance with stronger retrievers and broader boundary characterization, were not part of the original weaknesses, and we are happy to address them in the revision.
> > >
> > > Two brief clarifications:
> > >
> > > - **On stronger retrievers.** Full ElicitR is built on an already-initialized retriever (ElicitR-init), and generative regularization still delivers **substantial gains (+7.6/+7.3 over ElicitR-init** on BRIGHT at 1B/3B), confirming the method remains effective even when the retriever has been pre-trained on raw text.
> > >
> > > - **On broad applicability.** Across all four scales (0.1B–3B), ElicitR improves over RepLLaMA on **41/48 task-scale combinations (85%)** with an **average relative gain of 16–29%**, suggesting the method is broadly effective across diverse reasoning tasks.
> > >
> > > We appreciate the reviewer's recognition and encouragement and will address these points thoroughly in the revision.
> > >
> > > Best,
> > >
> > > Authors of ElicitR

---

### Official Review · Reviewer_yWeY · 2026-03-12

**Soundness:** 2
**Presentation:** 3
**Significance:** 3
**Originality:** 3
**Overall Recommendation:** 4
**Confidence:** 3

**Summary:**

This paper proposes ElicitR, a framework that augments standard contrastive retrieval training with a generative regularization objective. The authors observe that during contrastive fine-tuning, reasoning-intensive retrieval performance (like BRIGHT) peaks early and then declines, while general retrieval (BEIR) remains stable. To address this, ElicitR co-trains the retriever with a small 135M language model. During NTP, the LM predicts tokens in each batch element conditioned on other in-batch texts, weighted by retriever similarity scores. Gradients from NTP regularize the retriever to capture fine-grained cross-text dependencies beyond binary relevance. The framework is evaluated at 0.1B–3B scales. ElicitR improves BRIGHT by 16–29% over RepLLaMA while maintaining BEIR performance, achieving 23.1 nDCG@10 at 3B.

**Compliance With Llm Reviewing Policy:**

Affirmed.

**Final Justification:**

Thank the authors for your responses. I think my major concerns have been resolved. I have increased my score, but please do add these extensive results and examples in the revision.

**Key Questions For Authors:**

1. **Can you provide qualitative retrieval examples?** Showing cases where ElicitR succeeds and RepLLaMA fails would help ground the "reasoning" claim with concrete evidence.
2. **What characterizes the tasks where ElicitR degrades performance (Pony, AoPS at 3B)?** Is there a systematic pattern? Does the generative regularizer hurt on tasks requiring specific reasoning types (e.g., formal mathematics)?

**Limitations:**

Yes.

**Strengths And Weaknesses:**

**Strengths:**
1. The motivation is interesting and fair enough.
2. The paper is generally well-written with a clear narrative architecture.
3. The idea that reasoning capabilities are latent in LM-based retrievers and can be elicited through regularization rather than re-trained via synthetic data is a thought-provoking perspective. The consistent gains across four model scales support this hypothesis.

**Weaknesses:**
1. My biggest concern is that the paper claims to offer an "orthogonal alternative" to synthetic data approaches, yet does not compare with any of them. For example, ReasonIR-8B (Shao et al., 2025) and DIVER (Long et al., 2025). These methods are cited but not benchmarked. Without this comparison, the paper's central narrative that synthetic supervision is unnecessary is fundamentally undermined.
2. **Inconsistent per-task results are not discussed.** At the 3B scale, Pony drops from 9.3 to 3.0 (−68%) and AoPS drops from 14.8 to 12.8 (−13%). This pattern is consistent across scales for Pony. The paper does not acknowledge or explain these failures, which could reflect systematic limitations of the approach on certain reasoning types.

---

> ### Author Rebuttal · Authors · 2026-03-30
>
> We thank the reviewer for the thoughtful evaluation and for recognizing the interesting motivation, clear writing, and the novel insight that reasoning capabilities are latent in LM-based retrievers. We address each concern below.
> ## W1: Comparison with Synthetic Data Approaches
> **ElicitR-3B closely matches ReasonIR-8B on BRIGHT despite fundamentally different resource requirements**. ReasonIR-8B [1] achieves 24.4 nDCG@10 (as noted in our response to Reviewer oQHP); ElicitR-3B reaches 23.1, only 1.3 points behind, while using **2.6× fewer parameters** (3B vs. 8B) and **no synthetic data** (only **491K** MS MARCO pairs vs. ReasonIR's **1.7M** samples including ~345K synthesized by a Llama 70B model). Moreover, part of ReasonIR's training data is **synthesized directly from BRIGHT's corpus** documents, giving it a domain advantage. ElicitR uses no BRIGHT-related data whatsoever, making this small gap particularly notable.
>
> This comparison directly supports our central claim: substantial reasoning-intensive retrieval gains can be achieved by _eliciting_ latent reasoning through generative regularization, without the cost, pipeline complexity, or potential bias of synthetic supervision.
>
> **ElicitR is orthogonal and complementary to synthetic data approaches.** Our claim is that synthetic supervision is _not_ the only path for retriever learning. ElicitR addresses a fundamentally different bottleneck, i.e., contrastive overfitting,  and opens a complementary direction that synthetic data approaches do not address. We will clarify this framing in the revision.
>
> We note that DIVER [2] operates in a fundamentally different regime, a multi-stage pipeline with query expansion, hybrid retrieval, and LLM reranking, and is not directly comparable to a single-stage retriever.
>
> ## W2&Q2: Results on Pony, AoPS
> Thank you for pointing out this. The generative regularizer does **not** hurt formal reasoning tasks. Our ablation study (Table 11) reveals this drop is caused by the retriever's **initialization phase**, not the generative regularization mechanism itself.
>
> The key evidence is the **ER w. only init LM** setting, which pairs an initialized auxiliary LM with an _uninitialized_ retriever backbone. This isolates the effect of the regularization mechanism from the retriever initialization. Under this setting, Pony recovers to 11.1 and AoPS to 16.1, both **surpassing** RepLLaMA (9.3 and 14.8 respectively). This directly demonstrates that the generative regularization mechanism itself is **beneficial** for these tasks, and the degradation in the full setup stems solely from the retriever's initialization.
>
>
> In short, the mechanism is not the culprit. The root cause is a data-centric domain shift: our generic initialization corpus (e.g., Wikipedia) lacks the strict symbolic logic required for math theorems and exact code syntax, inadvertently shifting the retriever's latent space away from these niche domains, as evidenced by the extremely low ER-init scores (Pony: 2.4; AoPS: 9.6). Notably, this is a targeted weakness confined to two highly specialized tasks; across most of the remaining tasks, ElicitR generally and substantially outperforms RepLLaMA.
>
>
> In the revision, we plan to curate a broader, domain-specific initialization corpus to fully realize ElicitR's potential on these highly specialized tasks.
>
>
> ## Q1: Qualitative Retrieval Examples
> We thank the reviewer for this suggestion, which helps readers intuitively appreciate ElicitR's qualitative advantages.
>
> The Monterey temperature example (qid `45292` in MS MARCO) in Fig. 1(b) provides an intuition: under ElicitR, the retriever correctly distinguishes between NegX (topically relevant, weight 0.48) and NegY (irrelevant, weight 0.00), while InfoNCE collapses both to near-zero.
>
> We provide a further case from BRIGHT's **sustainable_living** task (qid=`54`): `How environmentally friendly is palm oil with a certified organic label?`
>
> RepLLaMA retrieves a doc about **RSPO** certification: `...Look for the RSPO label to ensure you purchase products made with certified sustainable palm oil…`. It shares _heavy_ lexical overlap with the query (`certified`, `palm oil`, `environmentally responsible`, overlap=62.5%) but answers about the **wrong certification system**.
>
> ElicitR retrieves the correct doc that directly addresses organic certification: `...organic labels do not establish standards against rainforest destruction and land grabbing`, requiring reasoning to distinguish **organic ≠ sustainable** despite _lower_ surface overlap (25.0%).
>
> This exemplifies how contrastive-only training traps the retriever in surface-level semantics matching: RepLLaMA cannot distinguish two certification systems sharing nearly identical vocabulary. ElicitR's generative regularization enables capturing this nuanced distinction.
>
> [1] ReasonIR: Training Retrievers for Reasoning Tasks, COLM 2025
>
> [2] DIVER: A Multi-Stage Approach for Reasoning-intensive Information Retrieval, arXiv:2508.07995

---

> > ### Author Rebuttal · Reviewer_yWeY · 2026-04-03
> >
> > Thank the authors for your responses. I think my major concerns have been resolved. I have increased my score, but please do add these extensive results and examples in the revision.

---

> > > ### Author Response · Authors · 2026-04-03
> > >
> > > Thank you for the positive reassessment and for the constructive feedback throughout the review process. We will incorporate all additional results and examples discussed during the rebuttal into the revision. We are grateful for the suggestions, which have helped strengthen the paper!
> > >
> > > Regards,
> > >
> > > Authors of ElicitR

---

### Official Review · Reviewer_Yyvj · 2026-03-16

**Soundness:** 2
**Presentation:** 2
**Significance:** 3
**Originality:** 2
**Overall Recommendation:** 3
**Confidence:** 5

**Summary:**

This paper studies reasoning-intensive retrieval and is built on a central hypothesis: the backbone of dense retrieval models may already encode substantial reasoning ability, but this ability is suppressed during contrastive training. Based on this view, the authors argue that, rather than learning reasoning from synthetic data, it may be more effective to elicit this latent capability through generative regularization. To this end, the paper proposes a joint model-training framework in which a dense retriever and a lightweight language model are optimized together, with the language model serving as a regularizer for dense retrieval training. Experimental results show that the proposed method improves performance on the reasoning-intensive BRIGHT benchmark while maintaining retrieval effectiveness on BEIR.

**Compliance With Llm Reviewing Policy:**

Affirmed.

**Final Justification:**

I appreciate the authors’ effort in providing a more detailed response. However, I remain unconvinced that the added analysis sufficiently supports the paper’s central claim that InfoNCE specifically harms reasoning-intensive retrieval and that generative regularization is particularly well suited to address this problem.

First, the early-peak phenomenon observed on BRIGHT remains largely unexplained. Since training is conducted on MS MARCO while evaluation is on BRIGHT, the degradation could reflect a fairly standard cross-dataset generalization issue, where continued fitting to the training distribution reduces transfer performance on an out-of-distribution test set. In its current form, the evidence does not convincingly show that this phenomenon is specific to reasoning-intensive retrieval or specifically caused by the limitations of InfoNCE.

Second, the additional findings on similarity distribution shift and calibration are not, by themselves, very compelling evidence for the paper’s core argument. It is already well known that neural retrieval models can be over-confident, so these observations do not yet provide a sufficiently novel or mechanistic explanation for why reasoning-intensive retrieval is uniquely affected.

Third, the comparison with entropy regularization is insufficient to establish the proposed method’s particular suitability. There are many other plausible and commonly used regularization strategies, especially teacher- or cross-encoder-based distillation methods that provide soft ranking signals for dense retrievers. Showing that entropy regularization is weak does not convincingly demonstrate that the proposed generative regularization is the most appropriate or most effective approach here.

On novelty, I agree that identifying a new research question or empirical phenomenon can itself be a meaningful contribution. However, in this paper, the case for novelty hinges on a convincing explanation of why generative regularization is particularly well-suited to reasoning-intensive retrieval. At present, that connection still feels underdeveloped. As a result, the work currently reads more like an application of Revela’s idea to the BRIGHT setting, followed by a post-hoc interpretation, rather than a sufficiently well-substantiated new contribution.

Overall, while I appreciate the authors’ thoughtful responses and the empirical improvements are interesting, I am still inclined to keep my original score of weak reject.

**Key Questions For Authors:**

- The proposed method appears closely related to Cai et al. (2025). Could the authors clarify the main technical differences and better position the novelty of this work?
- Could the authors provide more details on the implementation of generative regularization, especially how the passage-level attention in Equation 2 interacts with standard token-level Transformer attention?
- Figure 1(a) shows an early-peak phenomenon on BRIGHT under contrastive training. Do the authors have any explanation or analysis of why this happens?
- Why does generative regularization help reasoning-intensive retrieval in particular? Could similar improvements also be achieved by other forms of regularization, such as similarity regularization with teacher models?

**Limitations:**

Yes.

**Strengths And Weaknesses:**

Strengths:
- The paper identifies an early-peak phenomenon on the BRIGHT benchmark when dense retrieval models are trained with contrastive learning. This observation raises the interesting possibility that contrastive training for retrieval may impair performance on reasoning-intensive retrieval tasks. This is a novel and potentially important perspective.
- The proposed method yields a substantial performance gain on BRIGHT, and the improvement appears to be quite significant empirically.

Weaknesses:
- The proposed method appears highly similar to Cai et al. (2025). As a result, the paper’s novelty and its technical distinction from prior work need to be articulated more clearly.
- The description of the core method, generative regularization, is too brief, and several important technical details are left unspecified. For example, beyond the passage-level attention weights defined by retrieval similarity in Equation 2, it is unclear whether the generative model still uses the standard Transformer token-level attention. If so, the paper should clarify how token-level attention is integrated with passage-level attention during inference.
- Although Figure 1(a) presents the early-peak phenomenon, the paper does not further investigate the failure patterns and their underlying cause.
- While the experiments suggest that the proposed generative regularization effectively improves performance on reasoning-intensive retrieval tasks, the paper provides little theoretical explanation for why this mechanism works. It would also be valuable to discuss whether similar gains could be achieved through alternative forms of regularization, such as using a teacher model to regularize dense retrieval similarities.

---

> ### Author Rebuttal · Authors · 2026-03-30
>
> We thank the reviewer for recognizing our novel identification of contrastive overfitting and the substantial performance gains of ElicitR. We hope our replies address your concerns.
> ## Q1: Difference from Revela [1]
> Thanks for pointing it out, and we will make this more clear. ElicitR adapts the in-batch attention mechanism from Revela, as stated in Sec. 3.2.2. However, ElicitR's novelty is grounded in three distinct contributions that go well **beyond** the mechanism. (1) **Different problem**. Revela is a _self-supervised framework_ without query-doc pairs. ElicitR targets _supervised_ fine-tuning and mitigating contrastive overfitting, entirely out of Revela’s scope. (2) **Novel observation and solution**: We are the **first** to identify BRIGHT’s early-peak phenomenon during contrastive learning (Fig.1a), reframing reasoning-intensive retrieval away from data synthesis. (3) **Different functional role**: Revela uses in-batch attention as its **sole** objective on **homogeneous** raw texts; ElicitR uses it as an **auxiliary** regularizer on **query-positive-negative** batches alongside InfoNCE.
>
> Table 3 directly supports this: ElicitR-init (Revela's setup) achieves only 13.8/15.8 on BRIGHT (1B/3B), far below RepLLaMA (17.7/20.0), demonstrating Revela alone is insufficient.
> ## Q2: Implementation Details of Generative Regularization
>
> **On inference**. The in-batch attention is **training-time only** as a training regularization. At inference, ElicitR is a standard dual-encoder with **zero additional cost**, and the auxiliary LM is **not used** at all. We will make it more prominent in the revision.
>
> **On the layer-level interaction.** As detailed in App. A.2 (L689), the in-batch attention output is **directly added** to the standard causal self-attention output at each layer, before LayerNorm and FFN, with no additional normalization. This preserves the original residual stream structure and ensures in-batch attention acts as a pure additive correction. On the gradient side, $L_{genreg}$ flows through the in-batch attention weights $\alpha_{ij}$ back to the retriever $R_\theta$, while the LM $M_\phi$ receives gradients through both paths.
>
> Following your suggestion, we will make both points explicit with layer-level equations in the revision.
>
> ## Q3 & Q4: Early-Peak Cause + Why Generative Regularization Helps
> **The cause**. InfoNCE reduces every document to a binary label, discarding fine-grained relationships (L158-162). In Fig 1(b), for the query (qid `45292` in MSMARCO), NegX provides Monterey temperature data (semantically **closer** to the query) while NegY contains only indirect airport information, yet InfoNCE assigns both an identical near-zero signal. BEIR requires only lexical/semantic matching and remains stable; BRIGHT requires nuanced semantic relevance, precisely the signal that binary supervision erases. Please also refer to the examples in our response to Reviewer yWeY’s Q1.
>
> **Why generative regularization resolves this.** NTP forces the retriever to assign weights such that cross-text conditioning genuinely aids token prediction. In Fig.1(b), ElicitR assigns .52 to the positive and .48 to NegX but .00 to NegY, indicating that a fine-grained weighting is learned among texts. Three other analyses confirm this: ElicitR eliminates late-stage BRIGHT degradation (Fig. 2), shifts positive-probability toward lower confidence (Fig. 3), and reduces ECE across all backbones (Tab. 4).
>
> **On alternative regularizations.** We thank the reviewer for this insightful suggestion. We fully agree that teacher-model-guided regularization is well-motivated. We would point out, however, that **the auxiliary LM in ElicitR is itself functioning as a teacher model**: it teaches the retriever fine-grained relationships among all in-batch texts, between queries, positives, and negatives. Importantly, Revela already demonstrates that this NTP-based LM teaching signal substantially **outperforms REPLUG-style teacher distillation** [2] for training retrievers.
>
> To further validate whether broader regularization strategies could achieve similar gains through simpler means, we additionally study **entropy regularization** (Sec. 5.3(L403) and Table 17), which tests the effect of smoothing the binary similarity distribution without cross-text conditioning. It yields only small and inconsistent improvements across all scales and configurations (1B: 18.6 vs. ElicitR 21.4; 3B: 20.1 vs. 23.1), confirming that ElicitR’s benefit comes specifically from the structured, inter-text NTP signal rather than generic regularization.
>
> We believe ElicitR's contributions, i.e., a novel empirical observation, a principled diagnosis, and a validated solution, collectively justify its significance, and we are happy to discuss further.
>
> [1] Revela: Dense Retriever Learning via Language Modeling, ICLR 2026
>
> [2] REPLUG: Retrieval-Augmented Black-Box Language Models, NAACL 2024

---

> > ### Author Rebuttal · Reviewer_Yyvj · 2026-04-03
> >
> > Thank you for the responses. However, I do not think they fully resolve my concerns.
> >
> > On Q1, I still find the technical contribution somewhat limited relative to Cai et al. (2025), which already proposed the core generative regularization idea. In my reading, the main additional step in this paper is to combine that mechanism with supervised InfoNCE training and to show improved performance in this setting. While the empirical gains are interesting, this extension by itself does not yet strike me as sufficiently surprising or substantial from a novelty perspective.
> >
> > On Q3 and Q4, I do not think the current rebuttal provides enough evidence to support the paper’s central claims. A single case study is not sufficient to explain either why InfoNCE is fundamentally insufficient for reasoning-intensive retrieval or why generative regularization is particularly well suited to address this issue. Since the main technical mechanism is borrowed from Cai et al. (2025), the paper’s contribution depends heavily on the strength of this analysis. At the moment, I find the explanation too brief and not fully convincing.
> >
> > Overall, while I appreciate the rebuttal and the additional clarifications, I remain unconvinced that the paper’s technical and analytical contributions are strong enough for ICML.

---

> > > ### Author Response · Authors · 2026-04-03
> > >
> > > Thank you for the continued engagement and for clarifying your remaining concerns. Given the space constraints of the rebuttal, we were not able to provide the full depth of analysis that these questions deserve. We appreciate the opportunity to address them in more detail. We address Q3/Q4 first as the empirical evidence directly informs the novelty discussion in Q1.
> > >
> > > **On Q3/Q4: Sufficiency of the analysis.**
> > >
> > > We appreciate this concern and agree that the strength of the analysis is critical given the paper's contribution structure. We would like to clarify the scope of analytical evidence provided, which we believe extends beyond a single case study, though we fully acknowledge that further theoretical grounding would strengthen the work.
> > >
> > > Specifically, the paper and rebuttal provide **four independent and complementary lines of analytical evidence**:
> > >
> > > - **(1) Training dynamics (Fig. 2):** ElicitR eliminates the late-stage BRIGHT degradation observed in RepLLaMA training, providing a direct, checkpoint-level demonstration that generative regularization prevents the overfitting pattern, not just a post-hoc correlation.
> > >
> > > - **(2) Similarity distribution shift (Fig. 3):** ElicitR shifts the positive-document similarity distribution toward lower confidence compared to RepLLaMA, indicating that the retriever preserves nuanced query-document relationships rather than collapsing to binary separation.
> > >
> > > - **(3) Calibration improvement (Table 4):** ElicitR reduces Expected Calibration Error across all four backbones (135M–3B), providing an independent quantitative signal that the learned representations are better calibrated and less overfit to binary contrastive labels.
> > >
> > > - **(4) Entropy regularization ablation (Table 17):** This ablation directly tests the most natural alternative explanation, namely that generic confidence smoothing could account for the gains. It cannot: entropy regularization yields only small and inconsistent improvements (1B: 18.6 vs. ElicitR 21.4; 3B: 20.1 vs. 23.1), confirming that the gains are specific to the structured inter-text NTP signal.
> > >
> > > We believe these four analyses, taken together, provide systematic empirical support for the central claims. That said, we take your point that a deeper mechanistic or theoretical explanation would further strengthen the paper, and we are committed to expanding the analysis in the revision, for example by including probing experiments on representation geometry or a more formal connection between NTP regularization and representation diversity.
> > >
> > > **On Q1: Novelty relative to Revela.**
> > >
> > > With the above evidence in mind, we hope the following helps clarify why we view the contribution as going beyond combining Revela with InfoNCE.
> > >
> > > The core point is that **the research problem itself is new**. Revela operates in a self-supervised setting without query-document pairs and does not encounter contrastive overfitting. The central finding of this work, that contrastive fine-tuning systematically suppresses reasoning-intensive retrieval while leaving general retrieval largely unaffected, could not have emerged from Revela's setup. We believe that identifying this failure mode, diagnosing its cause, and demonstrating a targeted solution constitutes a scientific contribution in its own right, independent of the specific mechanism used to address it.
> > >
> > > Beyond the problem formulation, **the adaptation itself is non-trivial**. Revela operates on homogeneous raw-text chunks, whereas ElicitR applies in-batch attention to query-positive-negative batches, which is a **fundamentally different data structure** that required rethinking how cross-text conditioning interacts with contrastive supervision. **Table 3** provides systematic ablations confirming that the gains are not attributable to any single component in isolation: removing initialization, freezing the LM, or using only the initialized LM each leads to meaningful degradation. The performance emerges specifically from the interaction of all components during supervised fine-tuning.
> > >
> > > We recognize that reasonable reviewers can disagree on where the novelty bar lies. Our view is that the contribution follows a well-established pattern at top venues, i.e., **identifying a new phenomenon, diagnosing its cause, and validating a targeted solution**, and that the novelty is grounded in the problem and insight rather than solely in the mechanism. We note that this aligns with the original review's recognition of the early-peak phenomenon as "**a novel and potentially important perspective**" alongside "**substantial performance gains**," (which we really appreciate) and we hope the above elaboration helps clarify why these aspects represent a contribution that is distinct from Revela.
> > >
> > > Thank you again for the thoughtful engagement!
> > >
> > > Regards,
> > >
> > > Authors of ElicitR

---

### Decision · Program_Chairs · 2026-04-30

**Decision:**

Accept (regular)

**Comment:**

- The paper identifies a potentially important limitation of contrastive fine‑tuning for reasoning‑intensive retrieval and proposes a generative regularization framework that demonstrates consistent empirical gains on BRIGHT while maintaining general retrieval performance.
- The authors made a constructive effort during rebuttal to address concerns regarding task‑level degradation and comparisons with synthetic‑data‑based approaches. However, some concerns remain about novelty and analytical depth, as the proposed method builds on prior work such as Revela and primarily adapts in‑batch generative regularization to supervised InfoNCE training.
- Despite these limitations, the identification and empirical characterization of contrastive overfitting constitute a meaningful contribution that may stimulate further research.
- For these reasons, I recommend **Weak Accept**.